# Constant Acceleration Flow

**Dogyun Park**
Korea University
gg933@korea.ac.kr

**Sojin Lee**
Korea University
sojin_lee@korea.ac.kr

**Sihyeon Kim**
Korea University
sh_bs15@korea.ac.kr

**Taehoon Lee**
Korea University
98hoon@korea.ac.kr

**Youngjoon Hong**[*]
KAIST
hongyj@kaist.ac.kr

**Hyunwoo J. Kim**[*]
Korea University
hyunwoojkim@korea.ac.kr

## Abstract

Rectified flow and reflow procedures have significantly advanced fast generation by progressively straightening ordinary differential equation (ODE) flows. They operate under the assumption that image and noise pairs, known as couplings, can be approximated by straight trajectories with constant velocity. However, we observe that modeling with constant velocity and using reflow procedures have limitations in accurately learning straight trajectories between pairs, resulting in suboptimal performance in few-step generation. To address these limitations, we introduce Constant Acceleration Flow (CAF), a novel framework based on a simple constant acceleration equation. CAF introduces acceleration as an additional learnable variable, allowing for more expressive and accurate estimation of the ODE flow. Moreover, we propose two techniques to further improve estimation accuracy: initial velocity conditioning for the acceleration model and a reflow process for the initial velocity. Our comprehensive studies on toy datasets, CIFAR-10, and ImageNet 64×64 demonstrate that CAF outperforms state-of-the-art baselines for one-step generation. We also show that CAF dramatically improves few-step coupling preservation and inversion over Rectified flow. Code is available at https://github.com/mlvlab/CAF.

## 1 Introduction

Diffusion models [1, 2] learn the probability flow between a target data distribution and a simple Gaussian distribution through an iterative process. Starting from Gaussian noise, they gradually denoise to approximate the target distribution via a series of learned local transformations. Due to their superior generative capabilities compared to other models such as GANs and VAEs, diffusion models have become the go-to choice for high-quality image generation. However, their multi-step generation process entails slow generation and imposes a significant computational burden. To address this issue, two main approaches have been proposed: distillation models [3, 4, 5, 6, 7, 8, 9] and methods that simplify the flow trajectories [10, 11, 12, 13, 14] to achieve fewer-step generation. An example of the latter is *rectified flow* [10, 11, 13], which focuses on straightening ordinary differential equation (ODE) trajectories. Through repeated applications of the rectification process, called reflow, the trajectories become progressively straighter by addressing the *flow crossing* problem. Straighter flows reduce discretization errors, enabling fewer steps in the numerical solution and, thus, faster generation.

Rectified flow [10, 13] defines the straight ODE flow over time $t$ with a drift force $\mathbf{v}$, where each sample $\mathbf{x}_t$ transforms from $\mathbf{x}_0 \sim \pi_0$ to $\mathbf{x}_1 \sim \pi_1$ under a constant velocity $v = \mathbf{x}_1 - \mathbf{x}_0$. It

---

[*]Corresponding authors.

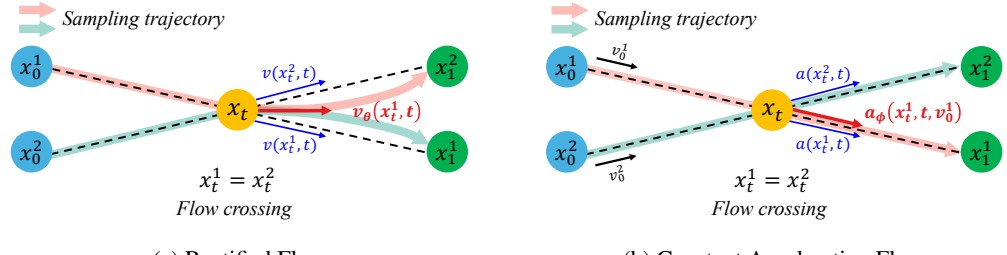

(a) Rectified Flow        (b) Constant Acceleration Flow

Figure 1: **Initial Velocity Conditioning (IVC).** We illustrate the importance of IVC to address the flow crossing problem, which hinders the learning of straight ODE trajectories during training. In Fig. 1a, Rectified flow suffers from approximation errors at the overlapping point $\mathbf{x}_t$ (where $\mathbf{x}_t^1 = \mathbf{x}_t^2$), resulting in curved sampling trajectories due to flow crossing. Conversely, Fig. 1b demonstrates that CAF, utilizing IVC, successfully estimates ground-truth trajectories by minimizing the ambiguity at $\mathbf{x}_t$.

approximates the underlying velocity $\mathbf{v}$ with a neural network $\mathbf{v}_\theta$. Then, it iteratively applies the reflow process to avoid flow crossing by rewiring the flow and building deterministic data coupling. However, constant velocity modeling may limit the expressiveness needed for approximating complex couplings between $\pi_0$ and $\pi_1$. This results in sampling trajectories that fail to converge optimally to the target distribution. Moreover, the interpolation paths after the reflow may still intersect—a phenomenon known as flow crossing—which leads to curved rectified flows because the model estimates different targets for the same input. As illustrated in Fig. 1a, instead of following the intended path from $\mathbf{x}_0^1$ to $\mathbf{x}_1^1$, a sampling trajectory from Rectified flow erroneously diverts towards $\mathbf{x}_1^2$ due to the flow crossing. Such flow crossing can make the accurate learning of straight ODE trajectories more challenging.

In this paper, we introduce the **C**onstant **A**cceleration **F**low (CAF), a novel ODE framework based on a constant acceleration equation, as outlined in (4). Our CAF generalizes Rectified flow by introducing acceleration as an additional learnable variable. This constant acceleration modeling offers the ability to control flow characteristics by manipulating the acceleration magnitude and enables a direct closed-form solution of the ODE, supporting precise and efficient sampling in just a few steps. Additionally, we propose two strategies to address the flow crossing problem. The first one is *initial velocity conditioning* (IVC) for the acceleration model, and the second one is to employ *reflow* to enhance the learning of initial velocity. Fig. 1b presents that CAF, with the proposed strategies, can accurately predict the ground-truth path from $\mathbf{x}_0^1$ to $\mathbf{x}_1^1$, even when flow crossing occurs. Through extensive experiments, from toy datasets to real-world image generation on CIFAR-10 [15] and ImageNet 64×64, we demonstrate that our CAF exhibits superior performance over Rectified flow and state-of-the-art baselines. Notably, CAF achieves superior Fréchet Inception Distance (FID) scores on CIFAR-10 and ImageNet 64×64 in conditional settings, recording FIDs of 1.39 and 1.69, respectively, thereby surpassing recent strong methods. Moreover, we show that CAF provides more accurate flow estimation than Rectified flow by assessing the 'straightness' and 'coupling preservation' of the learned ODE flow. CAF is also capable of few-step inversion, making it effective for real-world applications such as box inpainting.

To summarize, our contributions are as follows:

- We propose Constant Acceleration Flow (CAF), a novel ODE framework that integrates acceleration as a controllable variable, enhancing the precision of ODE flow estimation compared to the constant velocity framework.

- We propose two strategies to address the flow crossing problem: initial velocity conditioning for the acceleration model and a reflow procedure to improve initial velocity learning. These strategies ensure a more accurate trajectory estimation even in the presence of flow crossings.

- Through extensive experiments on synthetic and real datasets, CAF demonstrates remarkable performance, especially achieving the superior FID on CIFAR-10 and ImageNet 64×64 over strong baselines. We also demonstrate that CAF learns more accurate flow than Rectified flow by assessing the straightness, coupling preservation, and inversion.

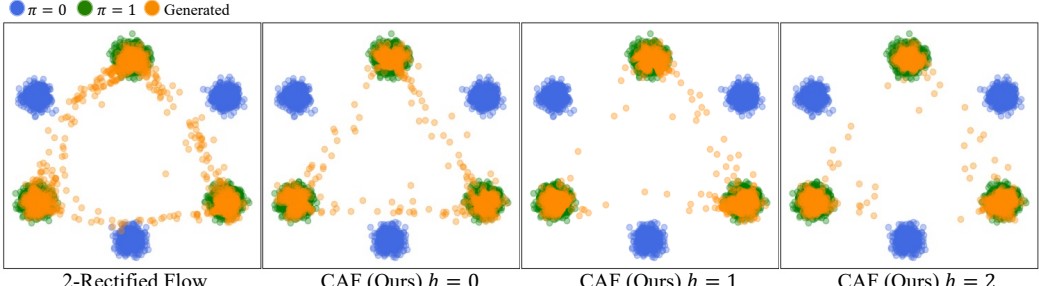

Figure 2: **2D synthetic dataset.** We compare results between 2-Rectified flow and our Constant Acceleration Flow (CAF) on 2D synthetic data. $\pi_0$ (blue) and $\pi_1$ (green) are source and target distributions parameterized by Gaussian mixture models. Here, the number of sampling steps is $N = 1$. While 2-Rectified flow frequently generates samples that deviate from $\pi_1$, CAF more accurately estimates the target distribution $\pi_1$. The generated samples (orange) from CAF form a more similar distribution as the target distribution $\pi_1$.

## 2  Related work

**Generative models.**  Learning generative models involves finding a nonlinear transformation between two distributions, typically denoted as $\pi_0$ and $\pi_1$, where $\pi_0$ is a simple distribution like a Gaussian, and $\pi_1$ is the complex data distribution. Various approaches have been developed to achieve this transformation. For example, variational autoencoders (VAE) [16, 17] optimize the Evidence Lower Bound (ELBO) to learn a nonlinear mapping from the latent space distribution $\pi_0$ to the data distribution $\pi_1$. Normalizing flows [18, 19, 20] construct a series of invertible and differentiable mappings to transform $\pi_0$ into $\pi_1$. Similarly, GANs [21, 22, 23, 24, 25] earn a generator that transforms $\pi_0$ into $\pi_1$ through an adversarial process involving a discriminator. These models typically perform a one-step generation from $\pi_0$ to $\pi_1$. In contrast, diffusion models [2, 26, 27, 28, 29, 30] propose learning the probability flow between the two distributions through an iterative process. This iterative process ensures stability and precision, as the model incrementally learns to reverse a diffusion process that adds noise to data. Diffusion models have demonstrated superior performance across various domains, including images [12, 31, 32, 33], 3D [34, 35, 36, 37], and video [38, 39, 40].

**Few-step diffusion models**  Addressing the slow generation speed of diffusion models has become a major focus in recent research: Distillation methods [3, 4, 5, 6, 7, 8, 9] seek to optimize the inference steps of pre-trained diffusion models by amortizing the integration of ODE flow. Consistency models [6, 7, 8] train a model to map any point on the pre-trained diffusion trajectory back to the data distribution, enabling fast generation. Rectified flow [10, 11, 13] is another direction, which focuses on straightening ODE trajectories under a constant velocity field. By straightening the flow and reducing path complexity, it allows for fast generation through efficient and accurate numerical solutions with fewer Euler steps. Recent methods such as AGM [41] also introduce acceleration modeling based on Stochastic Optimal Control (SOC) theory instead of relying solely on velocity. However, AGM predicts time-varying acceleration, which still requires multiple iterative steps to solve the differential equations. In contrast, our proposed CAF ODE assumes that the acceleration term is constant with respect to time. Therefore, there is no need to iteratively solve complex time-dependent differential equations. This simplification allows for a direct closed-form solution that supports efficient and accurate sampling in just a few steps.

## 3  Preliminary

**Rectified flow** [10, 13] is an ordinary differential equation-based framework for learning a mapping between two distributions $\pi_0$ and $\pi_1$. Typically, in image generation, $\pi_0$ is a simple tractable distribution, *e.g.*, the standard normal distribution, defined in the latent space and $\pi_1$ is the image distribution. Given empirical observations of $\mathbf{x}_0 \sim \pi_0$ and $\mathbf{x}_1 \sim \pi_1$ over time $t \in [0, 1]$, a flow is defined as

$$\frac{d\mathbf{x}_t}{dt} = \mathbf{v}(\mathbf{x}_t, t), \tag{1}$$

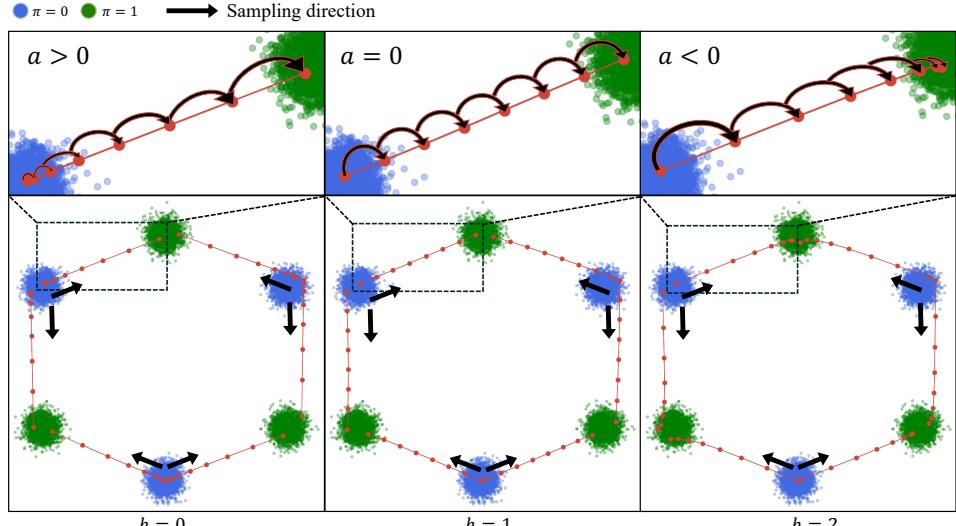

Figure 3: **Sampling trajectories of CAF with different $h$.** The sampling trajectories of CAF are displayed for different values of $h$, which determines the initial velocity and acceleration. $\pi_0$ and $\pi_1$ are mixtures of Gaussian distributions. We sample across sampling steps of $N = 7$ to show how sampling trajectories change with $h$.

where $\mathbf{x}_t = \mathcal{I}(\mathbf{x}_0, \mathbf{x}_1, t)$ is a time-differentiable interpolation between $\mathbf{x}_0$ and $\mathbf{x}_1$, and $\mathbf{v} : \mathbb{R}^d \times [0, 1] \to \mathbb{R}^d$ is a velocity field defined on data-time domain. Rectified flow learns the velocity field $\mathbf{v}$ with a neural network $\mathbf{v}_\theta$ by minimizing the following mean square objective:

$$\min_\theta \mathbb{E}_{\mathbf{x}_0, \mathbf{x}_1 \sim \gamma, t \sim p(t)} \left[ \|\mathbf{v}(\mathbf{x}_t, t) - \mathbf{v}_\theta(\mathbf{x}_t, t)\|^2 \right], \tag{2}$$

where $\gamma$ represents a coupling of $(\pi_0, \pi_1)$ and $p(t)$ is a time distribution defined on $[0, 1]$. The choice of interpolation $\mathcal{I}$ leads to various algorithms, such as Rectified flow [10], ADM [30], EDM [29], and LDM [42]. Specifically, Rectified flow proposes a simple linear interpolation between $\mathbf{x}_0$ and $\mathbf{x}_1$ as $\mathbf{x}_t = (1 - t)\mathbf{x}_0 + t\mathbf{x}_1$, which induces the velocity field $\mathbf{v}$ in the direction of $(\mathbf{x}_1 - \mathbf{x}_0)$, *i.e.*, $\mathbf{v}(\mathbf{x}_t, t) = \mathbf{x}_1 - \mathbf{x}_0$. This means the Rectified flow transports $\mathbf{x}_0$ to $\mathbf{x}_1$ along a straight trajectory with a *constant velocity*. After training $\mathbf{v}_\theta$, we can generate a sample $\mathbf{x}_1$ using off-the-shelf ODE solvers $\Phi$, such as the Euler method:

$$\mathbf{x}_{t+\Delta t} = \mathbf{x}_t + \Delta t \cdot \mathbf{v}_\theta(\mathbf{x}_t, t), \quad t \in \{0, \Delta t, \ldots, (N - 1) \cdot \Delta t\}, \tag{3}$$

where $\Delta t = \frac{1}{N}$ and $N$ is the total number of steps. To achieve faster generation with fewer steps without sacrificing accuracy, it is crucial to learn a straight ODE flow. Straight ODE flow minimize numerical errors encountered by the ODE solver.

**Reflow and flow crossing.** The trajectories of interpolants $\mathbf{x}_t$ may intersect—a phenomenon known as flow crossing—due to stochastic coupling between $\pi_0$ and $\pi_1$ (e.g., random pairing of $\mathbf{x}_0$ and $\mathbf{x}_1$). These intersections introduce approximation errors in the neural network, leading to curved sampling trajectories [10]. Our toy experiment, illustrated in Fig. 1a, clearly demonstrates this issue: the simulated sampling trajectories become curved due to flow crossing, rendering one-step simulation inaccurate. To address this problem, Rectified flow [10] introduces a reflow procedure. This procedure iteratively straightens the trajectories by reconstructing a more deterministic and direct pairing of $\mathbf{x}_0$ and $\mathbf{x}_1$ without altering the marginal distributions. Specifically, the reflow procedure involves generating a new coupling $\gamma$ of $(\mathbf{x}_0, \mathbf{x}_1 = \Phi(\mathbf{x}_0; \mathbf{v}_\theta^k))$ using a pre-trained Rectified flow model $\mathbf{v}_\theta^k$, where $k$ denotes the iteration of the reflow procedure, and $\Phi(\mathbf{x}_0; \mathbf{v}_\theta^k) = \mathbf{x}_0 + \int_0^1 \mathbf{v}_\theta^k(\mathbf{x}_t, t)dt$. By iteratively refining the coupling and the velocity field, the reflow procedure reduces flow crossing, resulting in straighter trajectories and improved accuracy in fewer steps.

# 4 Method

We aim to develop a generative model based on the ODE framework that enables faster generation without compromising quality. To achieve this, we propose a novel approach called **Constant Acceleration Flow** (CAF). Specifically, CAF formulates an ODE trajectory that transports $\mathbf{x}_t$ with a *constant acceleration*, offering a more expressive and precise estimation of the ODE flow compared to constant velocity models. Additionally, we propose two novel techniques that address the problem of flow crossing: 1) *initial velocity conditioning* and 2) *reflow procedure* for learning initial velocity. The overall training pipeline is presented in Alg. 1.

## 4.1 Constant Acceleration Flow

We propose a novel ODE framework based on the constant acceleration equation, which is driven by the empirical observations $\mathbf{x}_0 \sim \pi_0$ and $\mathbf{x}_1 \sim \pi_1$ over time $t \in [0, 1]$ as:

$$d\mathbf{x}_t = \mathbf{v}(\mathbf{x}_0, 0)dt + \mathbf{a}(\mathbf{x}_t, t)tdt, \tag{4}$$

where $\mathbf{v} : \mathbb{R}^d \times [0] \to \mathbb{R}^d$ is the initial velocity field and $\mathbf{a} : \mathbb{R}^d \times [0, 1] \to \mathbb{R}^d$ is the acceleration field. We abbreviate time variable $t$ for notation simplicity, *i.e.*, $\mathbf{v}(\mathbf{x}_0, 0) = \mathbf{v}(\mathbf{x}_0)$, $\mathbf{a}(\mathbf{x}_t, t) = \mathbf{a}(\mathbf{x}_t)$. By integrating both sides of (4) with respect to $t$ and assuming a constant acceleration field, *i.e.*, $\mathbf{a}(\mathbf{x}_{t_1}) = \mathbf{a}(\mathbf{x}_{t_2}), \forall t_1, t_2 \in [0, 1]$, we derive the following equation:

$$\mathbf{x}_t = \mathbf{x}_0 + \mathbf{v}(\mathbf{x}_0)t + \frac{1}{2}\mathbf{a}(\mathbf{x}_t)t^2. \tag{5}$$

Given the initial velocity field $\mathbf{v}$, the acceleration field $\mathbf{a}$ can be derived as

$$\mathbf{a}(\mathbf{x}_t) = 2(\mathbf{x}_1 - \mathbf{x}_0) - 2\mathbf{v}(\mathbf{x}_0), \tag{6}$$

by setting $t = 1$ and the constant acceleration assumption. Then, we propose a time-differentiable interpolation $\mathcal{I}$ as:

$$\mathbf{x}_t = \mathcal{I}(\mathbf{x}_0, \mathbf{x}_1, t, \mathbf{v}(\mathbf{x}_0)) = (1 - t^2)\mathbf{x}_0 + t^2\mathbf{x}_1 + \mathbf{v}(\mathbf{x}_0)(t - t^2), \tag{7}$$

by substituting (6) to (5). Using this result, we can easily simulate an intermediate sample $\mathbf{x}_t$ on our CAF ODE trajectory.

**Learning initial velocity field.** Selecting an appropriate initial velocity field is crucial, as different initial velocities lead to distinct flow dynamics. Here, we define the initial velocity field as a scaled displacement vector between $\mathbf{x}_1$ and $\mathbf{x}_0$:

$$\mathbf{v}(\mathbf{x}_0) = h(\mathbf{x}_1 - \mathbf{x}_0), \tag{8}$$

where $h \in \mathbb{R}$ is a hyperparameter that adjusts the scale of the initial velocity. This configuration enables straight ODE trajectories between distributions $\pi_0$ and $\pi_1$, similar to those in Rectified flow. However, varying $h$ changes the flow characteristics: 1) $h = 1$ simulates constant velocity flows, 2) $h < 1$ leads to a model with a positive acceleration, and 3) $h > 1$ results in a negative acceleration, as illustrated in Fig. 3. Empirically, we observe that the negative acceleration model is more effective for image sampling, possibly due to its ability to finely tune step sizes near data distribution.

The initial velocity field is learned using a neural network $\mathbf{v}_\theta$, which is optimized by minimizing the distance metric $d(\cdot, \cdot)$ between the target and estimated velocities as

$$\min_\theta \mathbb{E}_{\mathbf{x}_0, \mathbf{x}_1 \sim \gamma, t \sim p(t), \mathbf{x}_t \sim \mathcal{I}} \left[ d(\mathbf{v}(\mathbf{x}_0), \mathbf{v}_\theta(\mathbf{x}_t)) \right], \tag{9}$$

where $p(t)$ is a time distribution defined on $[0, 1]$. Note that our velocity model learns target initial velocity defined at $t = 0$. This differs from Rectified flow, which learns target velocity field defined over $t \in [0, 1]$.

**Learning acceleration field.** Under the assumption of constant acceleration, the acceleration field is derived from (6) as

$$\mathbf{a}(\mathbf{x}_t) = 2(\mathbf{x}_1 - \mathbf{x}_0) - 2\mathbf{v}(\mathbf{x}_0). \tag{10}$$

We learn the acceleration field using a neural network $\mathbf{a}_\phi$ by minimizing the distance metric $d(\cdot, \cdot)$ as:

$$\min_\phi \mathbb{E}_{\mathbf{x}_0, \mathbf{x}_1 \sim \gamma, t \sim p(t), \mathbf{x}_t \sim \mathcal{I}} \left[ d(\mathbf{a}(\mathbf{x}_t), \mathbf{a}_\phi(\mathbf{x}_t)) \right]. \tag{11}$$

In Sec. C, we theoretically show that CAF ODE preserves the marginal data distribution.

---

**Algorithm 1** Training process of Constant Acceleration Flow

---

**Require:** deterministic coupling $\gamma$, initial velocity scale $h$, $\mathbf{v}_\theta$, $\mathbf{a}_\phi$.
 1: **while** not converge **do**
 2:     $\mathbf{x}_0, \mathbf{x}_1 \sim \gamma, t \sim \text{Unif}([0, 1])$
 3:     $\mathbf{v}(\mathbf{x}_0) = h(\mathbf{x}_1 - \mathbf{x}_0)$                                     $\triangleright$ Target initial velocity
 4:     $\mathbf{x}_t = \mathcal{I}(\mathbf{x}_0, \mathbf{x}_1, t, \mathbf{v}(\mathbf{x}_0))$                                            $\triangleright$ Eq. (7)
 5:     $\mathcal{L}_{\text{vel}} = d(\mathbf{v}(\mathbf{x}_0), \mathbf{v}_\theta(\mathbf{x}_t))$
 6:     $\theta \leftarrow \theta - \nabla\mathcal{L}_{\text{vel}}$                                    $\triangleright$ update $\theta$ using SGD with gradient
 7: **end while**
 8: **while** not converge **do**
 9:     $\mathbf{x}_0, \mathbf{x}_1 \sim \gamma, t \sim \text{Unif}([0, 1]), \hat{\mathbf{v}}_\theta = \mathbf{v}_\theta(\mathbf{x}_0)$
10:     $\mathbf{a}(\mathbf{x}_t) = 2(\mathbf{x}_1 - \mathbf{x}_0) - 2\hat{\mathbf{v}}_\theta$                               $\triangleright$ Target acceleration
11:     $\mathbf{x}_t = \mathcal{I}(\mathbf{x}_0, \mathbf{x}_1, t, \hat{\mathbf{v}}_\theta)$                                           $\triangleright$ Eq. (7)
12:     $\mathcal{L}_{\text{acc}} = d(\text{sg}[\mathbf{a}(\mathbf{x}_t)], \mathbf{a}_\phi(\mathbf{x}_t, \hat{\mathbf{v}}_\theta))$
13:     $\phi \leftarrow \phi - \nabla\mathcal{L}_{\text{acc}}$                                    $\triangleright$ update $\phi$ using SGD with gradient
14: **end while**
15: **return** $\mathbf{v}_\theta, \mathbf{a}_\phi$

---

## 4.2 Addressing flow crossing

Rectified flow addresses the issue of flow crossing by a reflow procedure. However, even after the procedure, trajectories may still intersect each other. Such intersections hinder learning straight ODE trajectories, as demonstrated in Fig. 1a. Similarly, our acceleration model also encounters the flow crossing problem. This leads to inaccurate estimation, as the model struggles to predict estimation on these intersections correctly. To further address the flow crossing, we propose two techniques.

**Initial velocity conditioning (IVC).** We propose conditioning the estimated initial velocity $\hat{\mathbf{v}}_\theta = \mathbf{v}(\mathbf{x}_0)$ as the input of the acceleration model, $i.e.$, $\mathbf{a}_\phi(\mathbf{x}_t, \hat{\mathbf{v}}_\theta)$. This approach provides the acceleration model with auxiliary information on the flow direction, enhancing its capability to distinguish correct estimations and mitigate ambiguity at the intersections of trajectories, as illustrated in Fig. 1. Our IVC circumvents the non-intersecting condition required in Rectified flow (see Theorem 3.6 in [10]), which is a key assumption for achieving a straight coupling $\gamma$. By reducing the ambiguity arising from intersections, CAF can learn straight trajectories with less constrained couplings, which is quantitatively assessed in Tab. 4.

To incorporate IVC into learning the acceleration model, we reformulate (11) as:

$$\min_\phi \mathbb{E}_{\mathbf{x}_0, \mathbf{x}_1 \sim \gamma, t \sim p(t), \mathbf{x}_t \sim \mathcal{I}} \left[ d\left( \text{sg}[\mathbf{a}(\mathbf{x}_t)], \mathbf{a}_\phi(\mathbf{x}_t, \hat{\mathbf{v}}_\theta) \right) \right]. \tag{12}$$

where $\text{sg}[\cdot]$ indicates stop-gradient operation. Since our velocity model learns to predict the initial velocity (see (9)), we ensure that the model can handle both forward and reverse CAF ODEs, which start from $\mathbf{x}_0$ and $\mathbf{x}_1$, respectively. Thus, our acceleration model can generalize across different flow directions, enabling inversion as demonstrated in Sec. B.2.

**Reflow for initial velocity.** It is also important to improve the accuracy of the initial velocity model. Following [10], we address the inaccuracy caused by stochastic pairing of $\mathbf{x}_0$ and $\mathbf{x}_1$ by employing a pre-trained generative model $\psi$, which constructs a more deterministic coupling $\gamma$ of $\mathbf{x}_0$ and $\mathbf{x}_1$. We subsequently use this new coupling $\gamma$ to train the initial velocity and acceleration models.

## 4.3 Sampling

After training the initial velocity and acceleration models, we generate samples using the CAF ODE introduced in (4). The discrete sampling process is given by:

$$\mathbf{x}_{t+\Delta t} = \mathbf{x}_t + \Delta t \cdot \mathbf{v}_\theta(\mathbf{x}_0) + t' \cdot \Delta t \cdot \mathbf{a}_\phi(\mathbf{x}_t, t, \mathbf{v}_\theta(\mathbf{x}_0)), \tag{13}$$

where $N$ is the total number of steps, $\Delta t = \frac{1}{N}$, $t = i \cdot \Delta t$, and $t' = \frac{(2i+1)}{2} \cdot \Delta t$ where $i \in \{0, ..., N-1\}$ (See Alg. 2). We adopt $t'$ since it empirically improves accuracy, especially in the small $N$ regime. Notably, when $N = 1$ (one-step generation), $t'$ simplifies to $\frac{1}{2}$, leading to the closed-form solution in (5). See Alg. 3 for inversion algorithm.

---

**Algorithm 2** Sampling process of Constant Acceleration Flow

---

**Require:** velocity model $\mathbf{v}_\theta$, acceleration model $\mathbf{a}_\phi$, sampling steps $N$, $\pi_0$.

1: $\mathbf{x}_0 \sim \pi_0$
2: $\hat{\mathbf{v}}_\theta \leftarrow \mathbf{v}_\theta(\mathbf{x}_0)$
3: **for** $i = 0$ **to** $N - 1$ **do**
4:      $t \leftarrow \frac{i}{N}$
5:      $t' \leftarrow \frac{2i+1}{2N}$
6:      $\hat{\mathbf{a}}_\phi \leftarrow \mathbf{a}_\phi(\mathbf{x}_t, \mathbf{v}_\theta)$
7:      $\mathbf{x}_{t+\frac{1}{N}} \leftarrow \mathbf{x}_t + \frac{1}{N}\hat{\mathbf{v}}_\theta + \frac{t'}{N}\hat{\mathbf{a}}_\phi$
8: **end for**
9: **return** $\mathbf{x}_1$

---

## 5 Experiment

We evaluate the proposed Constant Acceleration Flow (CAF) across various scenarios, including both synthetic and real-world datasets. In Sec. 5.1, our investigation begins with a simple two-dimensional synthetic dataset, where we compare the performance of Rectified flow and CAF to clearly demonstrate the effectiveness of our model. Next, we extend our experiments to real-world image datasets, specifically CIFAR-10 (32×32) and ImageNet (64×64), in Sec. 5.2. These experiments highlight CAF's ability to generate high-quality images with a single sampling step. Furthermore, we conduct an in-depth analysis of CAF through evaluations of coupling preservation, straightness, inversion tasks, and an ablation study in Sec. 5.3.

### 5.1 Synthetic experiments

We demonstrate the advantages of the Constant Acceleration Flow (CAF) over the constant velocity flow model, Rectified Flow [10], through synthetic experiments. For the neural networks, we use multilayer perceptrons (MLPs) with five hidden layers and 128 units per layer. Initially, we train 1-Rectified flow on 2D synthetic data to establish a deterministic coupling. We then train both CAF and 2-Rectified flow. For CAF, we incorporate the initial velocity into the acceleration model by concatenating it with the input, ensuring that the model capacities of both CAF and 2-Rectified flow remain comparable. We set $d$ as $l_2$ distance. Fig. 2 presents samples generated from CAF in one step and from 2-Rectified flow in two steps. Our CAF more accurately approximates the target distribution $\pi_1$ than 2-Rectified flow. In particular, CAF with $h = 2$ (negative acceleration) learns the most accurate distribution. In contrast, 2-Rectified flow frequently generates samples that significantly deviate from $\pi_1$, indicating its difficulty in accurately estimating straight ODE trajectories. This experiment shows that reflowing alone may not overcome the flow crossing problem, leading to poor estimations, whereas our proposed acceleration modeling and IVC effectively address this issue. Moreover, Fig. 3 shows sampling trajectories from CAF trained with different hyperparameters $h$. It clearly demonstrates that $h$ controls the flow dynamics as we intended: $h > 1$ indicates negative acceleration, $h = 1$ represents constant velocity, and $h < 1$ corresponds to positive acceleration flows. Additional synthetic examples are provided in Fig. 6.

### 5.2 Real-data experiments

To further validate the effectiveness of our approach, we train CAF on real-world image datasets, specifically CIFAR-10 at 32×32 resolution and ImageNet at 64×64 resolution. To create a deterministic coupling $\gamma$, we utilize the pre-trained EDM models [29] and adopt the U-Net architecture of ADM [30] for the initial velocity and acceleration models. In the acceleration model, we double the input dimension of first layer to concatenate the initial velocity to the input $\mathbf{x}_t$ of the acceleration model, which marginally increases the total number of parameters. We set $h = 1.5$ and $d$ as LPIPS-Huber loss [43] for all real-data experiments.

**Baselines and evaluation.** We evaluate state-of-the-art diffusion models [1, 2, 7, 28, 29], GANs [22, 23, 24], and few-step generation approaches [6, 7]. We primarily assess the image generation quality of our method using the Fréchet Inception Distance (FID) [50] and Inception Score (IS) [51]. Additionally, we evaluate diversity using the recall metric following [6, 7, 10].

Table 1: Performance on CIFAR-10.

| Model | $N$ | Unconditional FID↓ | Conditional FID↓ |
|---|---|---|---|
| **GAN Models** | | | |
| BigGAN [22] | 1 | 8.51 | - |
| StyleGAN-Ada [23] | 1 | 2.92 | 2.42 |
| StyleGAN-XL [24] | 1 | - | 1.85 |
| **Diffusion/Consistency Models** | | | |
| Score SDE [1] | 2000 | 2.20 | - |
| DDPM [2] | 1000 | 3.17 | - |
| VDM [27] | 1000 | 7.41 | - |
| LSGM [28] | 138 | 2.10 | - |
| DDIM [26] | 10 | 13.36 | - |
| EDM [29] | 35 | 2.01 | 1.82 |
| | 5 | 37.75 | 35.54 |
| CT [6] | 2 | 5.83 | - |
| | 1 | 8.70 | - |
| **Diffusion/Consistency Models – Distillation** | | | |
| Diff-Instruct [9] | 1 | 4.53 | - |
| DMD [44] | 1 | 3.77 | - |
| DFNO [5] | 1 | 3.78 | - |
| TRACT [45] | 1 | 3.78 | - |
| KD [46] | 1 | 9.36 | - |
| CD [6] | 2 | 2.93 | - |
| | 1 | 3.55 | - |
| CTM [7] | 2 | 1.87 | 1.63 |
| | 1 | 1.98 | 1.73 |
| **Rectified Flow Models** | | | |
| 2-Rectified Flow [10] | 2 | 7.89 | 3.74 |
| | 1 | 11.81 | 6.88 |
| 2-Rectified Flow + Distill [10] | 1 | 4.84 | - |
| **CAF (Ours)** | 1 | 4.81 | 2.68 |
| **CAF + GAN (Ours)** | 1 | **1.48** | **1.39** |

Table 2: Performance on ImageNet $64 \times 64$.

| Model | $N$ | FID↓ | IS↑ | Rec↑ |
|---|---|---|---|---|
| **GAN Models** | | | | |
| BigGAN-deep [22] | 1 | 4.06 | - | 0.48 |
| StyleGAN-XL [24] | 1 | 2.09 | **82.35** | 0.52 |
| **Diffusion/Consistency Models** | | | | |
| DDIM [26] | 50 | 13.7 | - | 0.56 |
| | 10 | 18.3 | - | 0.49 |
| DDPM [2] | 250 | 11.0 | - | 0.58 |
| iDDPM [47] | 250 | 2.92 | - | 0.62 |
| ADM [30] | 250 | 2.07 | - | 0.63 |
| EDM [29] | 79 | 2.44 | 48.88 | **0.67** |
| | 5 | 55.3 | - | - |
| DPM-solver [48] | 20 | 3.42 | - | - |
| | 10 | 7.93 | - | - |
| DEIS [49] | 20 | 3.10 | - | - |
| | 10 | 6.65 | - | - |
| CT [6] | 2 | 11.1 | - | 0.56 |
| | 1 | 13.0 | - | 0.47 |
| **Diffusion/Consistency Models – Distillation** | | | | |
| Diff-Instruct [9] | 1 | 5.57 | - | - |
| DMD [44] | 1 | 2.62 | - | - |
| TRACT [45] | 1 | 7.43 | - | - |
| DFNO [5] | 1 | 7.83 | - | 0.61 |
| PD [3] | 1 | 15.39 | - | 0.62 |
| CD [6] | 2 | 4.70 | - | 0.64 |
| | 1 | 6.20 | 40.08 | 0.57 |
| CTM [7] | 2 | 1.73 | 64.29 | 0.57 |
| | 1 | 1.92 | 70.38 | 0.57 |
| **Rectified Flow Models** | | | | |
| **CAF (Ours)** | 1 | 6.52 | 37.45 | 0.62 |
| **CAF + GAN (Ours)** | 1 | **1.69** | 62.03 | 0.64 |

**Distillation.** Distilling a few-step student model from a pre-trained teacher model has recently become essential for high-quality few-step generation [6, 7, 10, 11]. InstaFlow [11] has observed that learning straighter trajectories and achieving good coupling significantly enhance distillation performance. Moreover, CTM [7] and DMD [44] incorporate an adversarial loss as an auxiliary loss to facilitate the training of the student model. We empirically found that incorporating the adversarial loss alone was sufficient to achieve superior performance for one-step sampling without introducing instability. For training details, please refer to Sec. A.

**CIFAR-10.** We present the experimental results on CIFAR-10 in Tab. 1. Our base unconditional CAF model (4.81 FID, $N = 1$) significantly improves the FID compared to recent state-of-the-art diffusion models (without distillation), including DDIM [26] (13.36 FID, $N = 10$), EDM (37.75 FID, $N = 5$), and 2-Rectified flow (7.89 FID, $N = 2$) in a few-step generation (*e.g.*, $N < 10$). We retrained 2-Rectified flow using the official codes of [10], achieving a slightly better performance than the officially reported performance (12.21 FID) for one-step generation [10]. CAF's remarkable 3.08 FID improvement over 2-Rectified flow ($N = 2$) highlights the effectiveness of acceleration modeling in a fast generation. Our approach is also effective in class-conditional generation, where the base CAF model (2.68 FID, $N = 1$) shows a significant FID improvement over EDM (35.54 FID, $N = 5$) and 2-Rectified flow (3.74 FID, $N = 2$). Additionally, after adversarial training, CAF achieves a superior FID of 1.48 for unconditional generation and 1.39 for conditional generation with $N = 1$. Lastly, we qualitatively compare the 2-Rectified flow and our CAF in Fig. 4, where CAF generates more vivid samples with intricate details than 2-Rectified flow.

**ImageNet.** We extend our evaluation to the ImageNet dataset at 64×64 resolution to demonstrate the scalability and effectiveness of our CAF model on more complex and higher-resolution images. Similar to the results on CIFAR-10, our base conditional CAF model significantly improves the FID compared to recent state-of-the-art diffusion models (without distillation) in the small $N$ regime (*e.g.*, $N < 10$). Specifically, CAF (6.52 FID, $N = 1$) outperforms models such as DPM-solver [48] (7.93 FID, $N = 10$), CT [6] (11.1 FID, $N = 2$), and EDM [29] (55.3 FID, $N = 5$). This validates that the superior performance of CAF can be effectively generalized to complex and large-scale datasets. Additionally, after adversarial training, CAF outperforms or is competitive with state-of-the-art distillation baselines in one-step generation. Notably, CAF achieves the best FID performance of 1.69, surpassing strong baselines. We also demonstrate one-step qualitative results in Fig. 14.

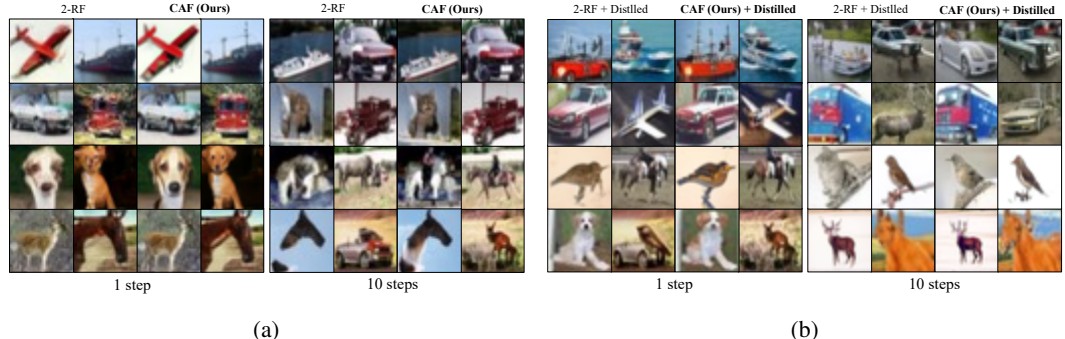

| 2-RF | CAF (Ours) | 2-RF | CAF (Ours) | 2-RF + Distlled | CAF (Ours) + Distilled | 2-RF + Distlled | CAF (Ours) + Distilled |

1 step | 10 steps | 1 step | 10 steps

(a) | (b)

Figure 4: **Qualitative results on CIFAR-10.** We compare the quality of generated images from 2-Rectified flow and CAF (Ours) with $N = 1$ and 10. Each image $\mathbf{x}_1$ is generated from the same $\mathbf{x}_0$ for both models. CAF generates more vivid images with intricate details than 2-RF for both $N$.

Table 3: Coupling preservation.

| Metric | 2-Rectified Flow | CAF (ours) |
|---|---|---|
| LPIPS $\downarrow$ | 0.092 | **0.041** |
| PSNR $\uparrow$ | 29.79 | **33.16** |

Table 4: Flow straightness comparison.

| Dataset | 2-Rectified Flow | CAF (ours) |
|---|---|---|
| 2D | 0.065 | **0.058** |
| CIFAR-10 | 0.043 | **0.034** |

Table 5: Ablation study on CIFAR-10 ($N = 1$).

| Config | Constant acceleration | $v_0$ condition | Reflow procedure | FID$\downarrow$ |
|---|---|---|---|---|
| A | ✗ | ✗ | ✗ | 378 |
| B | ✗ | ✗ | ✔ | 6.88 |
| C | ✔(h=1.5) | ✗ | ✔ | 3.82 |
| D | ✔(h=1.5) | ✔ | ✔ | **2.68** |
| E | ✔(h=1) | ✔ | ✔ | 3.02 |
| F | ✔(h=0.5) | ✔ | ✔ | 2.73 |

## 5.3 Analysis

**Coupling preservation.** We evaluate how accurately CAF and Rectified flow approximate the deterministic coupling obtained from pre-trained models via a reflow procedure. To analyze this, we first conduct synthetic experiments where the interpolant paths $\mathcal{I}$ are crossed, as illustrated in Fig. 5a. Due to the flow crossing, the sampling trajectory of Rectified flow fails to preserve the ground-truth coupling (interpolation path $\mathcal{I}$), leading to a curved sampling trajectory. In contrast, our CAF learns the straight interpolation paths by incorporating acceleration, demonstrating superior coupling preservation ability.

Moreover, we evaluate the coupling preservation ability on real data from CIFAR-10. We randomly sample 1K training pairs $(\mathbf{x}_0, \mathbf{x}_1)$ from the deterministic coupling $\gamma$ and measure the similarity between $\mathbf{x}_1$ and $\hat{\mathbf{x}}_1$, where $\hat{\mathbf{x}}_1$ is a generated sample from $\mathbf{x}_0$. In other words, we measure the distance between a ground truth image and a generated image corresponding to the same noise. If the coupling is well-preserved, the distance should be small. We use PSNR and LPIPS [52] as distance measures. The result in Tab. 3 demonstrates that CAF better preserves coupling. In terms of PSNR, CAF outperforms Rectified flow by 3.37. This is consistent with the qualitative result in Fig. 5b, where $\hat{\mathbf{x}}_1$ from CAF resembles more to $\mathbf{x}_1$ (ground truth) than $\hat{\mathbf{x}}_1$ from Rectified flow.

**Flow straightness.** To evaluate the straightness of learned trajectories, we introduce the Normalized Flow Straightness Score (NFSS). Similar to previous works [10, 11], we measure flow straightness $\mathcal{S}$ by the $L^2$distance between the normalized displacement vector $(\mathbf{x}_0 - \mathbf{x}_1)$ and the normalized velocity vector $\dot{\mathbf{x}}_t$ as below:

$$\mathcal{S} = \mathbb{E}_{\mathbf{x}_0, \mathbf{x}_1, t} \left[ \left\| \frac{\mathbf{x}_1 - \mathbf{x}_0}{\|\mathbf{x}_1 - \mathbf{x}_0\|_2} - \frac{\dot{\mathbf{x}}_t}{\|\dot{\mathbf{x}}_t\|_2} \right\|_2^2 \right]. \qquad (14)$$

Here, a smaller value of $\mathcal{S}$ indicates a *straighter* trajectory. We compare $\mathcal{S}$ between CAF and Rectified flow using synthetic and real-world datasets, as presented in Tab. 4. For Rectified flow, we use $\dot{\mathbf{x}}_t = \mathbf{v}_\theta(\mathbf{x}_t)$, while for CAF, we use $\dot{\mathbf{x}}_t = \mathbf{v}_\theta(\mathbf{x}_0) + \mathbf{a}_\phi(\mathbf{x}_t)t$. The results show that CAF outperforms Rectified flow in flow straightness.

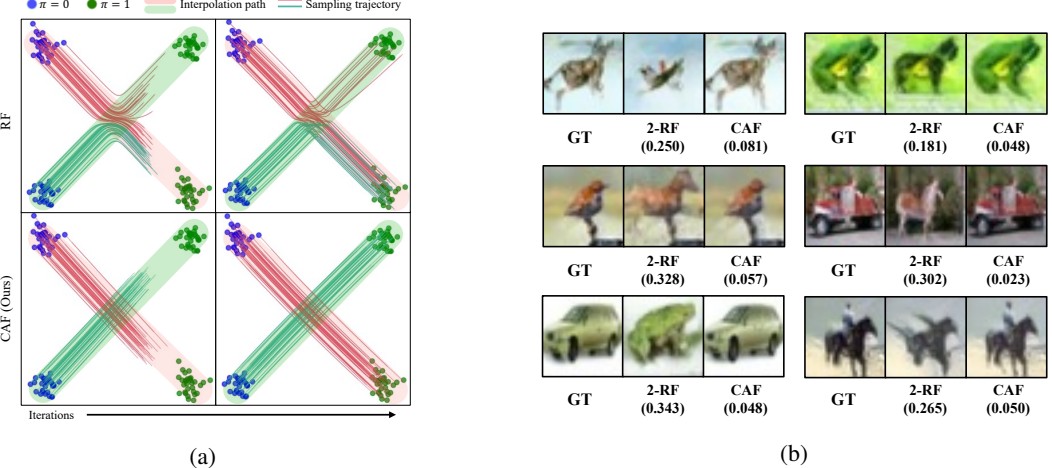

(a)                                                                        (b)

Figure 5: **Experiments for coupling preservation.** (a) We plot the sampling trajectories during training where their interpolation paths $\mathcal{I}$ are crossed. Due to the flow crossing, RF (top) *rewires* the coupling, whereas CAF (bottom) *preserves* the coupling of training data. (b) CAF accurately generates target images from the given noise (*e.g.*, a car from the car noise), while RF often fails (*e.g.*, a frog from the car noise). LPIPS [52] values are in parentheses.

**Inversion** We further demonstrate CAF's capability in real-world applications by conducting zero-shot tasks such as reconstruction and box inpainting using inversion. We provide implemenetation details and algorithms in Sec. B.2. As shown in the Tab. 6 and 7, our method achieves lower reconstruction errors (CAF: 46.68 PSNR vs. RF: 33.34 PSNR) and better zero-shot inpainting capabilities even with fewer steps compared to baselines. These improvements are attributed to CAF's superior coupling preservation capability. Moreover, we present qualitative comparisons between CAF and the baselines in Fig. 12 and 13, which further validates the quantitative results.

**Ablation study.** We conduct an ablation study to evaluate the effectiveness of components in our framework under the one-step generation setting ($N = 1$). We examine the improvements achieved by **1)** constant acceleration modeling, **2)** initial velocity ($\mathbf{v}_0$) conditioning, and **3)** the reflow procedure for $\mathbf{v}_0$. The configurations and results are outlined in Tab. 5. Specifically, A and B correspond to 1-Rectified flow and 2-Rectified flow, respectively. Configurations C to F represent our CAF frameworks, with C being our CAF without IVC. By comparing A,B,C, and F, we demonstrate that all three components in our framework substantially improve the performance. In addition, we analyze the final model across various acceleration scales controlled by $h$. The performance difference between D and F is relatively small, indicating that our framework is robust to the choice of hyperparameters. Empirically, we observe that configuration F, *i.e.*, CAF ($h = 1.5$) with negative acceleration, achieves the best FID of 2.68. Notably, our CAF without $\mathbf{v}_0$ conditioning, still outperforms rectified flow (configuration B) by 3.06 FID. This highlights the critical role of *constant acceleration modeling* in enhancing the quality of few-step generation. Also, we verify the significance of reflowing by comparing configurations A and B, which achieve 378 FID and 6.88 FID, respectively.

## 6   Conclusion

In this paper, we have introduced the Constant Acceleration Flow (CAF) framework, which enhances precise ODE trajectory estimation by incorporating a controllable acceleration variable into the ODE framework. To address the flow crossing problem, we proposed two strategies: initial velocity conditioning and a reflow procedure. Our experiments on toy datasets, real-world dataset demonstrate CAF's capabilities and scalability, achieving state-of-the-art FID scores. Furthermore, we conducted extensive ablation studies and analyses—including assessments of flow straightness, coupling preservation, and real-world applications—to validate and deepen our understanding of the effectiveness of our proposed components in learning accurate ODE trajectories. We believe that CAF offers a promising direction for efficient and accurate generative modeling, and we look forward to exploring its applications in more diverse settings such as 3D and video.

# Acknowledgement

This work was supported by ICT Creative Consilience Program through the Institute of Information & Communications Technology Planning & Evaluation (IITP) grant funded by the Korea government (MSIT) (IITP-2024-RS-2020-II201819, 10%), the National Research Foundation of Korea (NRF) grant funded by the Korea government (MSIT) (NRF-2023R1A2C2005373, 45%), and the Virtual Engineering Platform Project (Grant No. P0022336, 45%), funded by the Ministry of Trade, Industry & Energy (MoTIE, South Korea).

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

# A  Implementation details

We utilize the pre-trained EDM model [29] to build the deterministic coupling $\gamma$ for training our models. To construct deterministic couplings for CIFAR-10 and ImageNet, we select $N = 18$ and $N = 40$, respectively, using deterministic sampling following the protocol in [29]. For CIFAR-10 and ImageNet, we generate 1M and 3M pairs, respectively. We use the batch size of 2048 and train for 700K/700K iterations on ImageNet. For CIFAR-10, we use the batch size of 512 and train for 500K/500K iterations. For all experiments, we use AdamW [53] optimizer with a learning rate of 0.0001 and apply an Exponential Moving Average (EMA) with a 0.999 decay rate. For training acceleration model, we initialize it with initial velocity model for faster convergence.

For adversarial training, we employ adversarial loss $\mathcal{L}_{\text{gan}}$ using real data $\mathbf{x}_{1,\text{real}}$ from [24]:

$$\mathcal{L}_{\text{gan},\eta}(\phi) = \mathbb{E}_{\mathbf{x}_{1,\text{real}}} \left[\log d_\eta(\mathbf{x}_{1,\text{real}})\right] + \mathbb{E}_{\mathbf{x}_0} \left[\log(1 - d_\eta(\hat{\mathbf{x}}_1))\right], \tag{15}$$

where $d_\eta$ is a discriminator and $\hat{\mathbf{x}}_1 = \mathbf{x}_0 + v_\theta(\mathbf{x}_0) + \frac{1}{2}a_\phi(\mathbf{x}_0, \mathbf{v}_\theta(\mathbf{x}_0))$. In the end, we use the following combined loss to update the acceleration model:

$$\mathcal{L}(\phi, \eta) = \mathcal{L}_{\text{acc}}(\phi) + \lambda_{\text{gan}}\mathcal{L}_{\text{gan}}(\phi, \eta), \tag{16}$$

where $\mathcal{L}_{\text{acc}}$ corresponds to (12) and $\lambda$ are weight hyperparameters. Following [42, 54], we employ adaptive weighting as $\lambda_{\text{gan}} = \frac{\|\nabla_{\phi_l}\mathcal{L}_{\text{acc}}(\phi)\|}{\|\nabla_{\phi_l}\mathcal{L}_{\text{gan}}(\phi,\eta)\|}$, where $\phi_l$ is the last layer of the acceleration model. Without $\mathcal{L}_{\text{acc}}$, we found the training unstable and frequently exhibit mode collapse issue, which is a common problem with adversarial training. We follow the training configuration from StyleGAN-XL [24]. We bilinearly upscale the image to 224×224 resolution and use EfficientNet [55] and DeiT-base [56] for extracting features. During the adversarial training, we only optimize the acceleration model and discriminator with the learning rate of 2e-5 and 1e-3, respectively. We keep the parameters of the initial velocity model fixed for stable training. The total training takes about 21 days with 8 NVIDIA A100 GPUs for ImageNet, and takes 10 days 8 NVIDIA RTX3090 GPUs for CIFAR-10.

# B  Additional results

## B.1  Additional qualitative results

**2D toy dataset.**    In Fig. 6, we provide additional generation results and sampling trajectories on various 2D synthetic datasets with $N = 1$, demonstrating the effectiveness of our approach for fast generation. Fig. 7 provides additional examples of coupling preservation on 2-RF and CAF.

**Real-world dataset.**    In Fig. 8 and 9, we show additional generation results from our base CAF model on CIFAR-10 with $N = 1, 10$, and 50. In Fig. 10, we compare the generation result between 2-RF and CAF distilled versions. Fig. 11 shows sampling results from our base CAF models with different hyperparameters $h$. Lastly, Fig. 14 shows the generation results on ImageNet with $N = 1$.

## B.2  Real-world applications

Inversion techniques are essential for real-world applications such as image and video editing [57, 58]. However, existing methods typically require 25–100 steps for accurate inversion, which can be computationally intensive. In contrast, our method significantly reduces the inference time by enabling inversion in just a few steps (*e.g.*, $N < 20$). We demonstrate this efficiency in two tasks: reconstruction and box inpainting.

To reconstruct $\mathbf{x}_1$, we first invert $\mathbf{x}_1$ to obtain $\hat{\mathbf{x}}_0$, as described in Alg. 3. We then use the generation process (Alg. 2) with $\hat{\mathbf{x}}_0$ and same initial velocity $\mathbf{v}_\theta(\mathbf{x}_1)$ used in Alg. 3 to generate $\hat{\mathbf{x}}_1$. For box inpainting, we inject conditional information—the non-masked image region—into the iterative inversion and generation procedures, as detailed in Alg. 4. As demonstrated in Tab. 6 and 7, our method achieves better reconstruction quality (CAF: 46.68 PSNR vs. RF: 33.34 PSNR) and zero-shot inpainting capability even with fewer steps compared to baseline methods. Qualitative results are presented in Fig. 12 and 13, which further illustrate the effectiveness of our approach. This demonstrate that our method can be efficiently used for real-world applications, offering both speed and accuracy advantages over existing techniques.

---

**Algorithm 3** Inversion process of Constant Acceleration Flow

---

**Require:** velocity model $\mathbf{v}_\theta$, acceleration model $\mathbf{a}_\phi$, sampling steps $N$, $\pi_1$.
1: $\mathbf{x}_1 \sim \pi_1$
2: $\hat{\mathbf{v}}_\theta \leftarrow \mathbf{v}_\theta(\mathbf{x}_1)$
3: **for** $i = N$ **to** 1 **do**
4:      $t \leftarrow \frac{i}{N}$
5:      $t' \leftarrow \frac{2i-1}{2N}$
6:      $\hat{\mathbf{a}}_\phi \leftarrow \mathbf{a}_\phi(\mathbf{x}_t, \hat{\mathbf{v}}_\theta)$
7:      $\mathbf{x}_{t-\frac{1}{N}} \leftarrow \mathbf{x}_t - \frac{1}{N}\hat{\mathbf{v}}_\theta - \frac{t'}{N}\hat{\mathbf{a}}_\phi$
8: **end for**
9: **return** $\mathbf{x}_0$

---

---

**Algorithm 4** Box inpainting of Constant Acceleration Flow

---

**Require:** velocity model $\mathbf{v}_\theta$, acceleration model $\mathbf{a}_\phi$, sampling steps $N$, reference image $\bar{\mathbf{x}}_1$, binary image mask $\Omega$ where 1 indicates the missing pixels.
1: $\sigma \sim \mathcal{N}(0, I)$
2: $\bar{\mathbf{x}} \leftarrow \bar{\mathbf{x}}_1 \odot (1 - \Omega) + \sigma \odot \Omega$            ▷ Create image with missing pixels and add noise $\sigma$
3: $\hat{\mathbf{v}}_\theta \leftarrow \mathbf{v}_\theta(\bar{\mathbf{x}})$
4: **for** $i = N$ **to** 1 **do**            ▷ Inversion steps
5:      $t \leftarrow \frac{i}{N}, t' \leftarrow \frac{2i-1}{2N}$
6:      $\hat{\mathbf{a}}_\phi \leftarrow \mathbf{a}_\phi(\mathbf{x}_t, \hat{\mathbf{v}}_\theta)$
7:      $\mathbf{x}_{t-\frac{1}{N}} \leftarrow \mathbf{x}_t - \frac{1}{N}\hat{\mathbf{v}}_\theta - \frac{t'}{N}\hat{\mathbf{a}}_\phi$
8:      $\mathbf{x}_{t-\frac{1}{N}} \leftarrow \mathbf{x}_{t-\frac{1}{N}} \odot (1 - \Omega) + (1-t)\sigma \odot \Omega, \;\; \sigma \sim \mathcal{N}(0, I)$
9: **end for**
10: $\hat{\mathbf{v}}_\theta \leftarrow \mathbf{v}_\theta(\mathbf{x}_0)$
11: **for** $j = 0$ **to** $N-1$ **do**            ▷ Generation steps
12:      $t \leftarrow \frac{j}{N}, t' \leftarrow \frac{2j+1}{2N}$
13:      $\hat{\mathbf{a}}_\phi \leftarrow \mathbf{a}_\phi(\mathbf{x}_t, \hat{\mathbf{v}}_\theta)$
14:      $\mathbf{x}_{t+\frac{1}{N}} \leftarrow \mathbf{x}_t + \frac{1}{N}\hat{\mathbf{v}}_\theta + \frac{t'}{N}\hat{\mathbf{a}}_\phi$
15:      $\mathbf{x}_{t+\frac{1}{N}} \leftarrow \bar{\mathbf{x}}_1 \odot (1 - \Omega) + \mathbf{x}_{t+\frac{1}{N}} \odot \Omega$
16: **end for**
17: **return** inpainted image $\mathbf{x}_1$

---

### B.3 Comparison with previous acceleration modeling literatures

Here, we elaborate on the crucial differences between AGM [41] and CAF. The main distinction is that CAF assumes constant acceleration, whereas AGM predicts time-dependent acceleration. Since the CAF ODE assumes that the acceleration term is constant with time, there is no need to solve time-dependent differential equations iteratively. This allows for a closed-form solution that supports efficient and accurate sampling, given that the learned velocity and acceleration models are accurate. Specifically, the solution for CAF ODE is given by:

$$\mathbf{x}_1 = \mathbf{x}_0 + \int_0^1 \mathbf{v}(\mathbf{x}_0) + \mathbf{a}(\mathbf{x}_t) \cdot t \, dt = \mathbf{x}_0 + \mathbf{v}(\mathbf{x}_0) + \int_0^1 \mathbf{a}(\mathbf{x}_t) \cdot t \, dt \tag{17}$$

$$= \mathbf{x}_0 + \mathbf{v}(\mathbf{x}_0) + \mathbf{a}(\mathbf{x}_t) \int_0^1 t \, dt = \mathbf{x}_0 + \mathbf{v}(\mathbf{x}_0) + \frac{1}{2}\mathbf{a}(\mathbf{x}_t) \tag{18}$$

The integral simplifies thanks to the constant acceleration assumption, leading to one-step sampling. In contrast, AGM's acceleration is time-varying, meaning that the differential equation cannot be reduced in an analytic form. It requires multiple steps to approximate the true solution accurately. In Tab. 8, we systemically compare AGM with our CAF, where CAF consistently outperforms AGM. Moreover, we conducted additional experiments where AGM was trained with deterministic couplings as in our reflow setting. Incorporating reflow into AGM did not improve its performance in the few-step regime, which further highlights the distinct advantage of CAF over AGM.

Table 6: Reconstruction error.

| Model | $N$ | PSNR ↑ | LPIPS ↓ |
|---|---|---|---|
| CM | - | N/A | N/A |
| CTM | - | N/A | N/A |
| EDM | 4 | 13.85 | 0.447 |
| 2-RF | 2 | 33.34 | 0.094 |
| 2-RF | 1 | 29.33 | 0.204 |
| **CAF (Ours)** | 1 | **46.68** | **0.007** |
| **CAF (+GAN) (Ours)** | 1 | 40.84 | 0.028 |

Table 7: Box inpainting.

| Model | NFE | FID ↓ |
|---|---|---|
| CM | 18 | 13.16 |
| CTM | - | N/A |
| EDM | - | N/A |
| 2-RF | 12 | 16.41 |
| **CAF (Ours)** | 12 | **10.39** |
| **CAF (+GAN) (Ours)** | 12 | 10.91 |

Table 8: Comparison between AGM and CAF.

| Model | Acceleration | Closed-form solution | Reflow for velocity | FID on CIFAR-10 ↓ |
|---|---|---|---|---|
| AGM [41] | Time-varying | No | No | 11.88 ($N = 5$) |
| AGM (enhanced ver.) | Time-varying | No | Yes | 15.23 ($N = 5$) |
| **CAF (Ours)** | Constant | Yes | Yes | **4.81** ($N = 1$) |

## C  Marginal preserving property of Constant Acceleration Flow

We demonstrate that the flow generated by our Constant Acceleration Flow (CAF) ordinary differential equation (ODE) maintains the marginal of the data distribution, as established by the definitions and theorem in [10].

**Definition C.1.** *For a path-wise continuously differentiable process* $\mathbf{x} = \{\mathbf{x}_t : t \in [0, 1]\}$*, we define its expected velocity* $\mathbf{v}^{\mathbf{x}}$ *and acceleration* $\mathbf{a}^{\mathbf{x}}$ *as follow:*

$$\mathbf{v}^{\mathbf{x}}(x, t) = \mathbb{E}\left[\frac{d\mathbf{x}_t}{dt} \mid \mathbf{x}_t = x\right], \quad \mathbf{a}^{\mathbf{x}}(x, t) = \mathbb{E}\left[\frac{d^2\mathbf{x}_t}{dt^2} \mid \mathbf{x}_t = x\right], \quad \forall x \in \text{supp}(\mathbf{x}_t). \quad (19)$$

*For* $x \notin supp(\mathbf{x}_t)$*, the conditional expectation is not defined and we set* $\mathbf{v}^{\mathbf{x}}$ *and* $\mathbf{a}^{\mathbf{x}}$ *arbitrarily, for example* $\mathbf{v}^{\mathbf{x}}(x, t) = 0$ *and* $\mathbf{a}^{\mathbf{x}}(x, t) = 0$*.*

**Definition C.2.** [10] *We denote that* $\mathbf{x}$ *is rectifiable if* $\mathbf{v}^{\mathbf{x}}$ *is locally bounded and the solution to the integral equation of the form*

$$\mathbf{z}_t = \mathbf{z}_0 + \int_0^t \mathbf{v}^{\mathbf{x}}(\mathbf{z}_t, t)dt, \quad \forall t \in [0, 1], \quad \mathbf{z}_0 = \mathbf{x}_0, \quad (20)$$

*exists and is unique. In this case,* $\mathbf{z} = \{\mathbf{z}_t : t \in [0, 1]\}$ *is called the rectified flow induced by* $\mathbf{x}$*.*

**Theorem 1.** [10] *Assume* $\mathbf{x}$ *is rectifiable and* $\mathbf{z}$ *is its rectified flow. Then* $\text{Law}(\mathbf{z}_t) = \text{Law}(\mathbf{x}_t), \forall t \in [0, 1]$*.*

Refer to [10] for the proof of Theorem 1.

We will now show that our CAF ODE satisfies Theorem 1 by proving that our proposed ODE (4) induces $\mathbf{z}$, which is the rectified flow as defined in Definition C.2. In (4), we define the CAF ODE as

$$\frac{d\mathbf{x}_t}{dt} = \frac{d\mathbf{x}_t}{dt}\bigg|_{t=0} + \frac{d^2\mathbf{x}_t}{dt^2} \cdot t. \quad (21)$$

By taking the conditional expectation on both sides, we obtain

$$\mathbf{v}^{\mathbf{x}}(x, t) = \mathbf{v}^{\mathbf{x}}(x, 0) + \mathbf{a}^{\mathbf{x}}(x, t) \cdot t, \quad (22)$$

from Definition C.1. Then, the solution of the integral equation of CAF ODE is identical to the solution in Definition C.2 by (22):

$$\mathbf{z}_t = \mathbf{z}_0 + \int_0^t \mathbf{v}^{\mathbf{x}}(\mathbf{z}_0, 0) + \mathbf{a}^{\mathbf{x}}(\mathbf{z}_t, t) \cdot t dt \quad (23)$$

$$= \mathbf{z}_0 + \int_0^t \mathbf{v}^{\mathbf{x}}(\mathbf{z}_t, t)dt. \quad (24)$$

This indicates that $\mathbf{z}$ induced by CAF ODE is also a rectified flow. Therefore, the CAF ODE satisfies the marginal preserving property, i.e., $\text{Law}(\mathbf{z}_t) = \text{Law}(\mathbf{x}_t)$, as stated in Theorem 1.

# D   Limitation and Broader impacts

## D.1   Limitations

One limitation of our model is the increased number of function evaluations (NFE) required for $N$-step generation. While Rectified flow achieves an NFE of $N$ by only computing the velocity at each step, our method necessitates an additional computation, resulting in a total NFE of $N + 1$. This is because we compute the initial velocity at the beginning and the acceleration at each step. Although this extra evaluation slightly increases the computational burden, it is relatively minor in terms of overall performance and still enables efficient few-step generation. Moreover, this additional step can be reduced by jointly predicting velocity and acceleration terms with a single model, which we leave for future work. Another limitation is the additional effort required to generate supplementary data. We utilize generated data to create a deterministic coupling of noise and data samples for training CAF. While generating more data enhances our model's performance, it can increase GPU usage, leading to higher carbon emissions.

## D.2   Broader Impacts

Recent advancements in generative models hold significant potential for societal benefits across a wide array of applications, such as image and video generation and editing, medical imaging analysis, molecular design, and audio synthesis. Our CAF framework contributes to enhancing the efficiency and performance of existing diffusion models, offering promising directions for positive impacts across multiple domains. This suggests that in practical applications, users can utilize generative models more rapidly and accurately, enabling a broad spectrum of activities. However, it is crucial to acknowledge potential risks that must be carefully managed. The increased accessibility of generative models also broadens the potential for misuse. As these technologies become more widespread, the possibility of their exploitation for fraudulent activities, privacy breaches, and criminal behavior increases. It is vital to ensure their ethical and responsible use to prevent negative impacts. Establishing regulated ethical standards for developing and deploying generative AI technologies is necessary to prevent such misuse. Additionally, imposing restricted access protocols or verification systems to trace and authenticate generated contents will help ensure their responsible use.

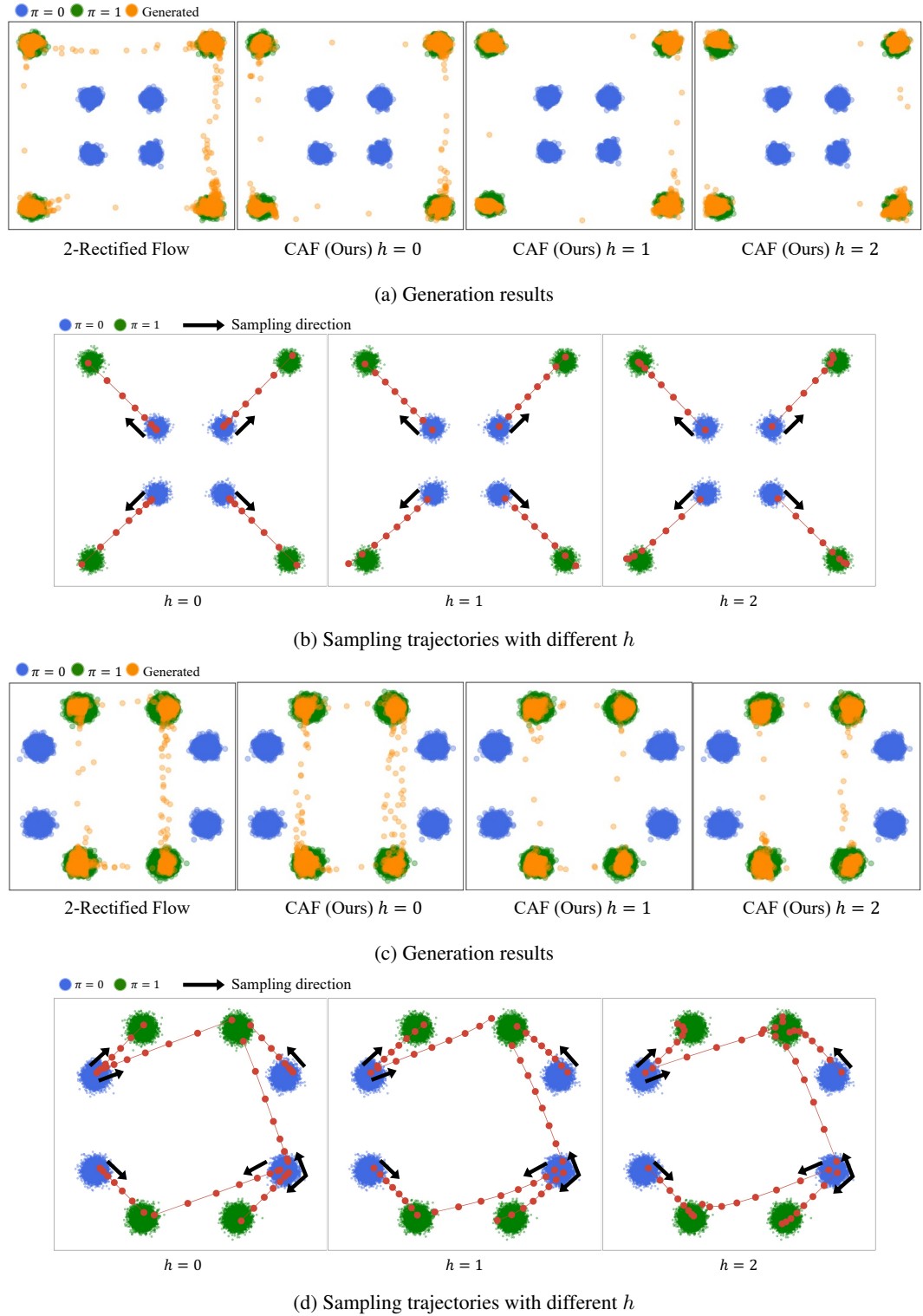

Figure 6: **Experiments on various 2D synthetic dataset.** We compare results between 2-Rectified Flow and our Constant Acceleration Flow (CAF) on 2D synthetic data. $\pi_0$ (blue) and $\pi_1$ (green) are source and target distributions parameterized by Gaussian mixture models. The generated samples (orange) from CAF form a more similar distribution as the target distribution $\pi_1$.

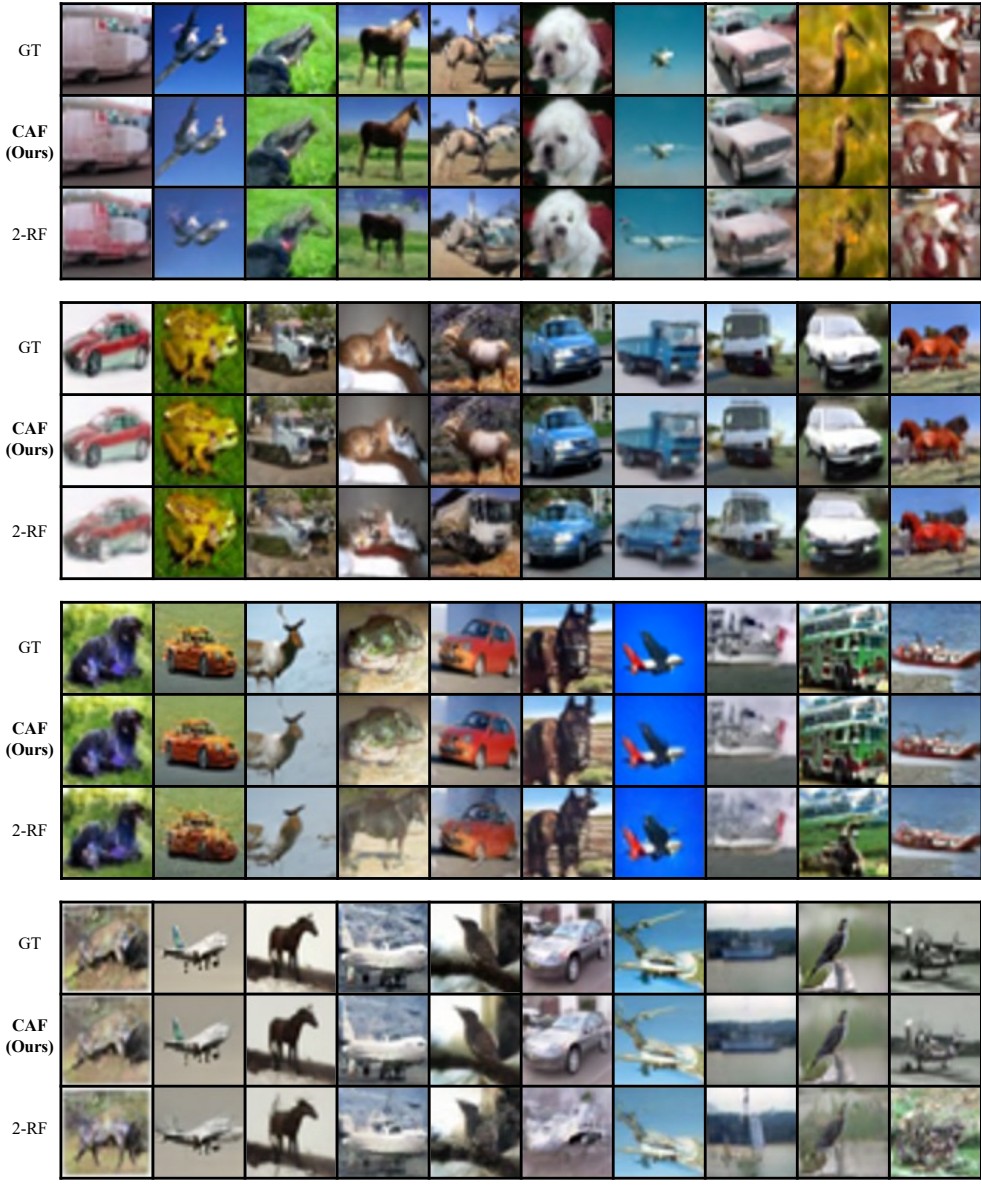

Figure 7: **Additional visualizations of coupling preservation on CIFAR-10.** CAF accurately generates target images ($x_1$) from the given noise ($x_0$), while Rectified Flow often fails to preserve coupling of $x_0$ and $x_1$ .

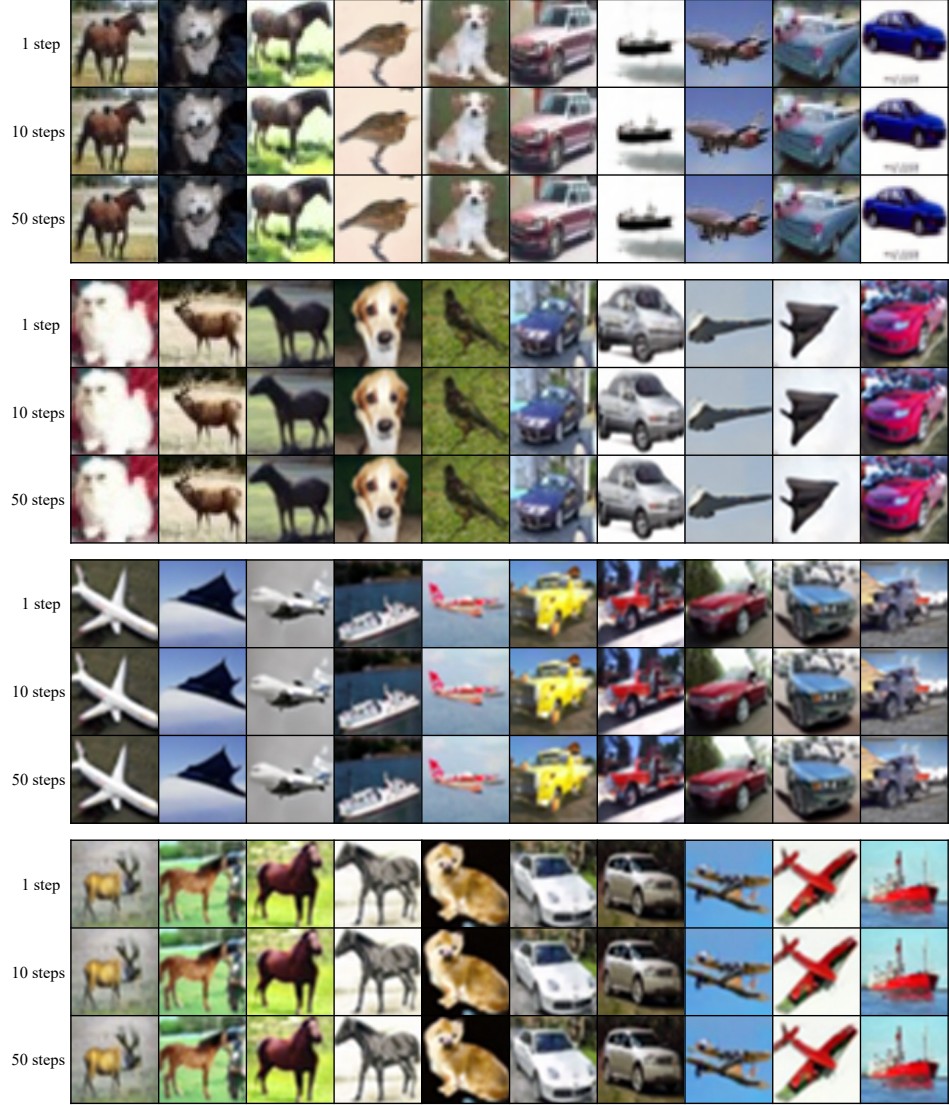

Figure 8: **Qualitative results on unconditional generation (CIFAR-10).** We illustrate generating images with varying sampling steps, demonstrating consistency quality even for a one-step generation.

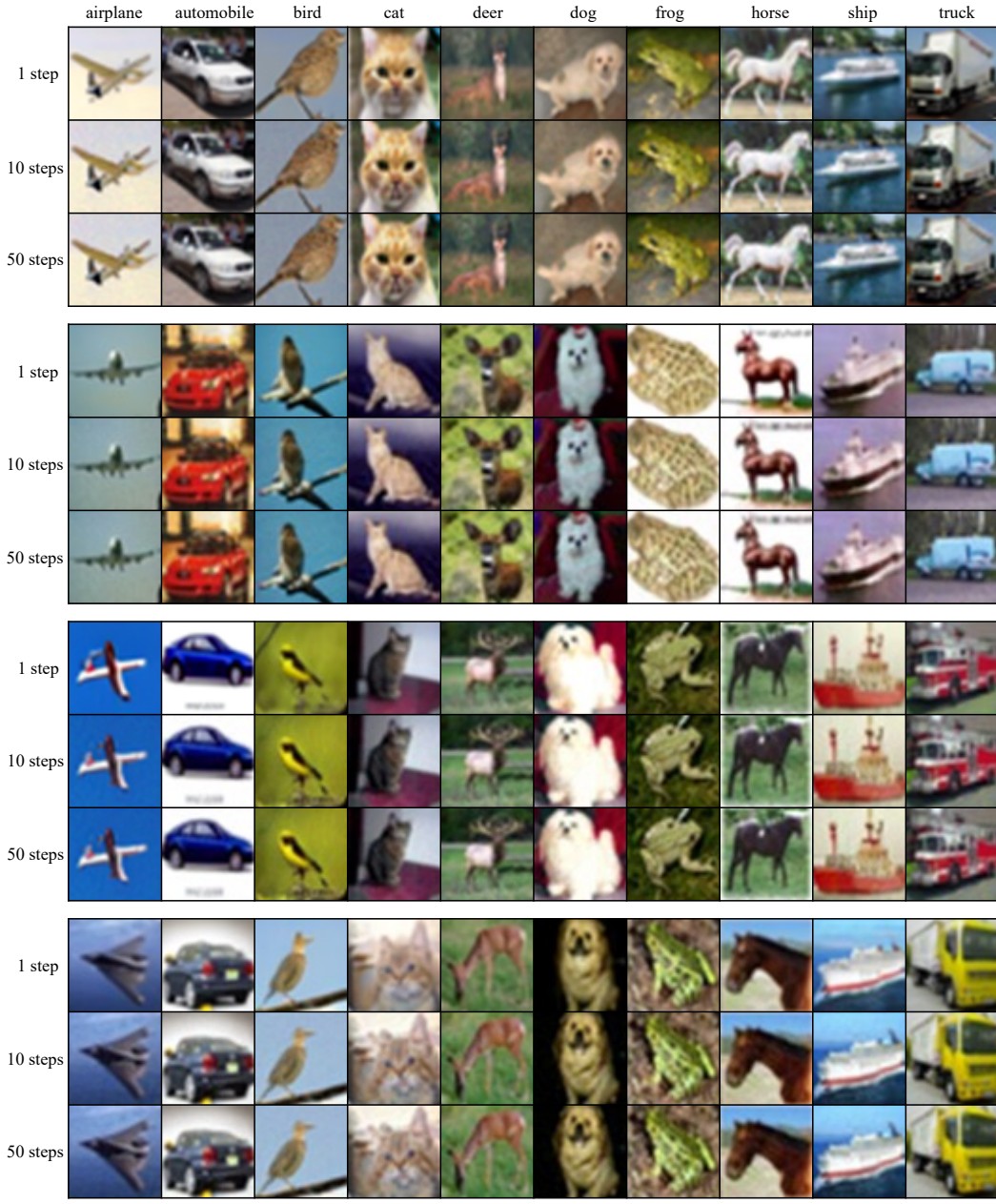

Figure 9: **Qualitative results on conditional generation (CIFAR-10).** We illustrate generating images with varying sampling steps, demonstrating consistency quality even for a one-step generation.

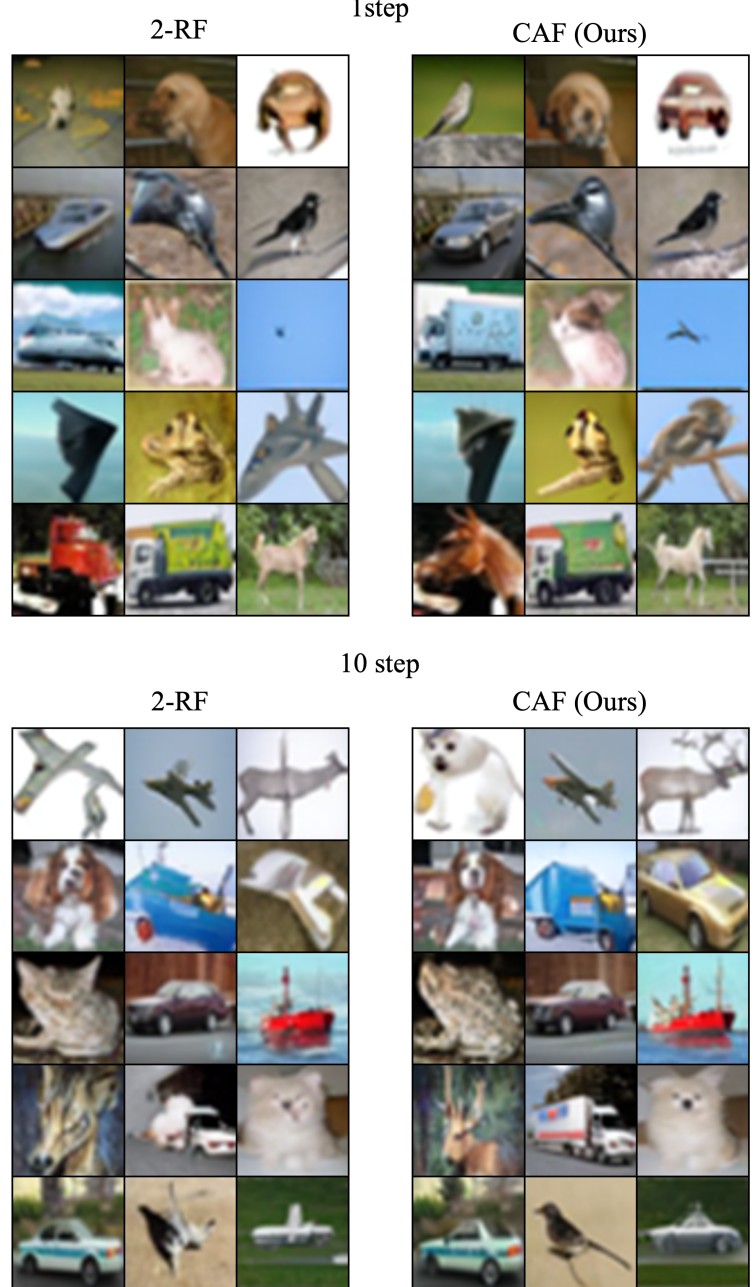

Figure 10: **Comparisons on unconditional generation (CIFAR-10).** We compare distilled model from 2-Rectified Flow (2-RF+Distill+GAN) and CAF (CAF+Distill+GAN) with qualitative results.

1 step

h = 1 h = 2

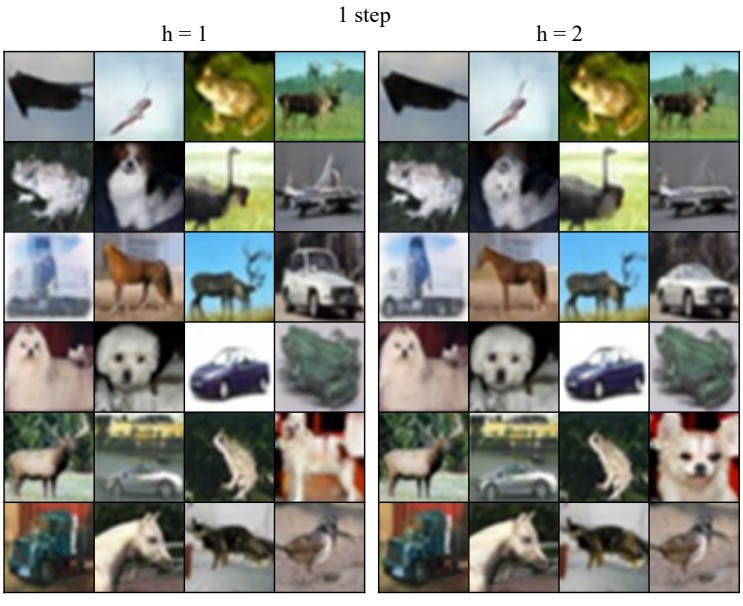

10 steps

h = 1 h = 2

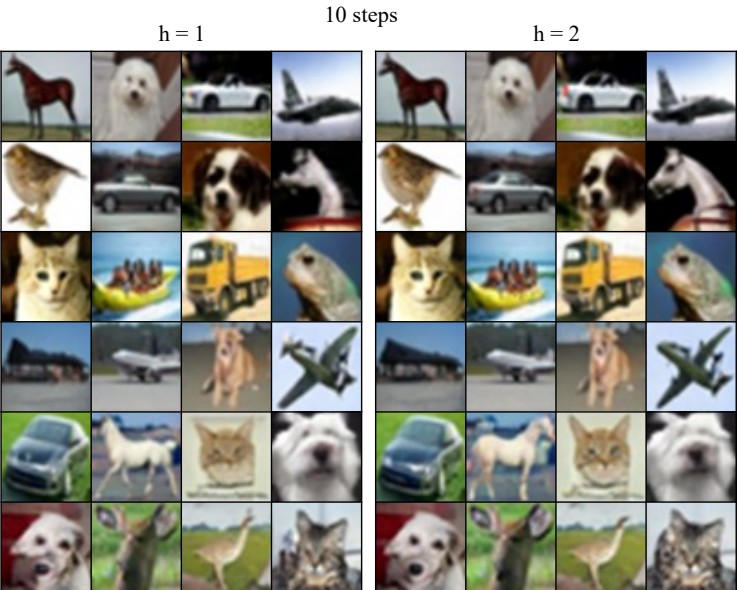

Figure 11: **Unconditional generation for different** $h$ **on CIFAR-10.** We display qualitative results of CAF for different values of $h$, indicating that our framework is robust to the choice of $h$.

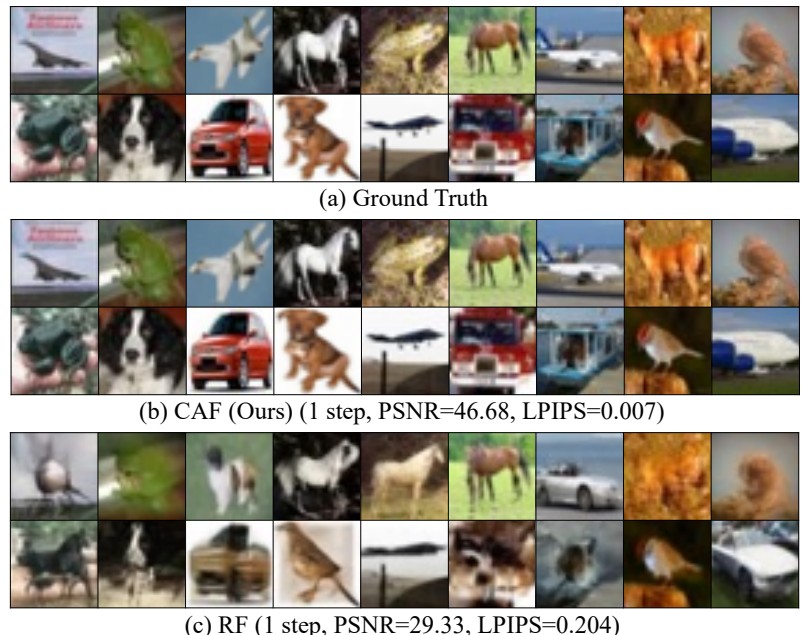

(a) Ground Truth

(b) CAF (Ours) (1 step, PSNR=46.68, LPIPS=0.007)

(c) RF (1 step, PSNR=29.33, LPIPS=0.204)

Figure 12: **Reconstruction results using inversion.**

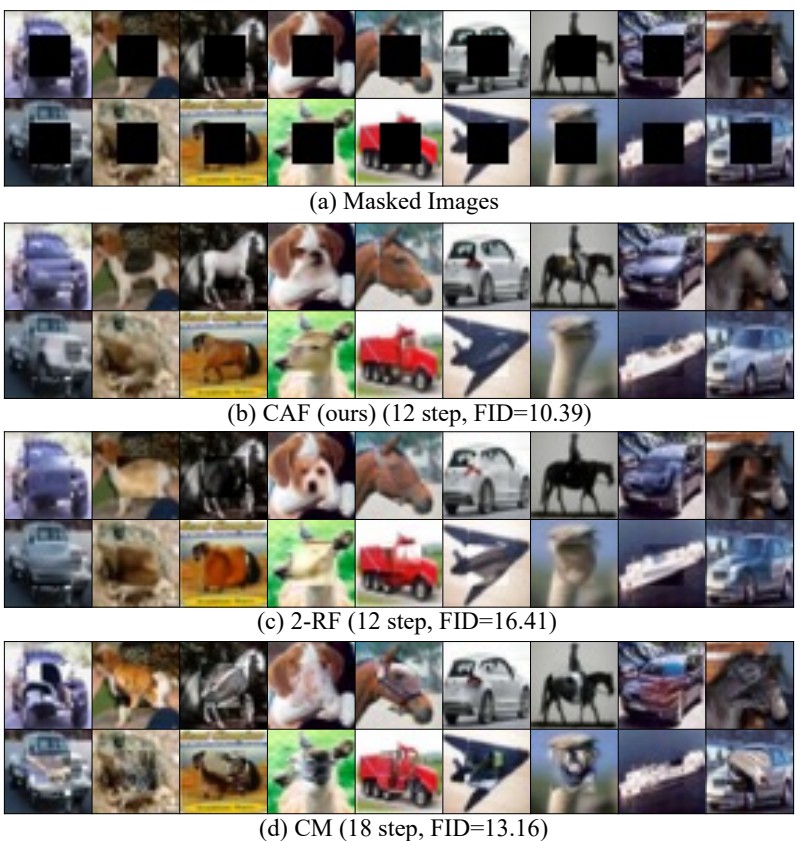

(a) Masked Images

(b) CAF (ours) (12 step, FID=10.39)

(c) 2-RF (12 step, FID=16.41)

(d) CM (18 step, FID=13.16)

Figure 13: **Zero-shot box inpainting results.** We use a 16×16 size mask for masked images in (a).
For consistency model in (d), we followed their official code for inpainting.

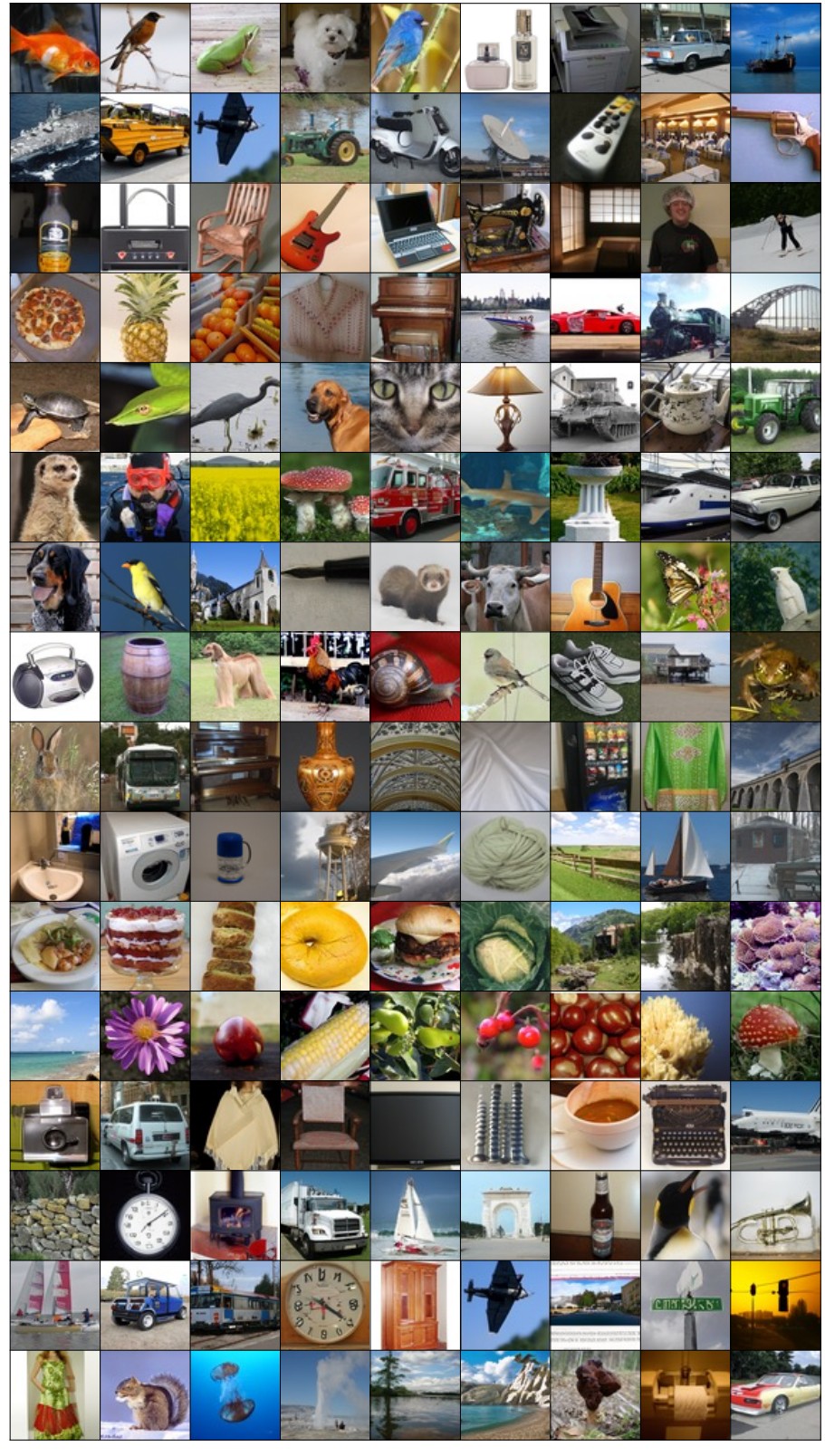

Figure 14: Qualitative results on conditional generation for ImageNet $64\times64$ ($N = 1$, FID=1.69).

