# OpenReview forum: "Constant Acceleration Flow"
_NeurIPS.cc/2024/Conference — NeurIPS 2024 poster_

### Official Review · Reviewer_oUYA · 2024-07-02

**Soundness:** 3
**Presentation:** 3
**Contribution:** 3
**Rating:** 6
**Confidence:** 3

**Summary:**

The paper presents an extension to rectified flow by Liu et al., 2023, a method based on optimal transport that can match samples from two distributions. This class of methods may have different uses but is often employed for the efficient generation of new samples and for morphing one sample to another. As such rectified flow and the extension proposed in this paper may be seen as finding efficient trajectories for use during generation by diffusion models. The method solves the transport flow in a simple Euler approximation of the ODE describing the flow. The paper identifies crossing of flow trajectories as a major obstacle in generating efficient paths in rectified flow and proposes constant acceleration flow (CAF). The paper proposes Initial Velocity Conditioning to produce the initialize the acceleration field in solving the ODE. Rectified flow uses constant velocity to solve the ODE using a reflow procedure to optimize the paths in a second step. Similarly, the proposed CAF also uses reflow to improve the initial paths. The paper reports experimental evaluation of the model with synthetic distributions and with CIFAR-10. The experiments demonstrate improved flow between distributions for synthetic data and improved FID score on CIFAR-10. The paper also reports an increase in straightness of trajectories.

**Strengths:**

The constant acceleration flow is novel to the best of my knowledge.

The paper describes the Initial Velocity Conditioning and related steps to make the approach work on CiFAR-10. The approach is suitably summarized in pseudo-code in two algorithms.

The synthetic experiments demonstrate the paper's motivation visually.

The ODE framework for the transport and the forward Euler solver makes the steps in the procedure easy to follow. The analogy makes it easy to see that constant velocity and constant acceleration lead to straight trajectories is the initial coupling of the samples x_0 and x_1 is correct.

A comparison with many alternative methods is provided in Table 1.

The proposed CAF visually leads to improved quality and stability on CIFAR-10 over rectified flow.

**Weaknesses:**

The paper presents arguably a substantial improvement of results obtainable with rectified flow in terms of quality at the price of increased computation. The paper does not attempt any argument why this is significant, i.e., has the improved quality the potential to lead to wider adaptation of rectified flow type methods. The paper has no real-world use case to motivate the work. The claim that experiments with CIFAR-10 at 32x32 pixels is a real-world example seems far-fetched.

The initialization of the method and its impact on the result is not well explored. The initial velocity calculation requires sample x_1 drawn from the target distribution. Both, the paper under consideration and rectified flow employ a pre-trained generative model to obtain that sample. The quality of initial samples may impact the results of CAF. The authors should provide evidence to the contrary or provide an experimental evaluation of the influence.

There is no theoretical justification why constant acceleration is superior to constant velocity flow generation. While the experimental evaluation appears clear, it remains unclear if this behavior is due to the tasks considered or holds more broadly.

The paper claims reflow as a contribution but it has been proposed by rectified flow. The same is true for the measurement of straightness of flow trajectory with Normalized Flow Straightness Score (NFSS) in Eqn 12.

The paper does not discuss how CIFAR-10 is used. One can speculate based on earlier work such as GLOW that unconditional generation and conditional generation may refer to use of class conditioning but a concise description of the experimental setup is required.

The use of a pre-trained generative model should also be taken into account in the required computational effort during training. It may be application-dependent if such a model is available or if it need to be trained.

**Questions:**

A discussion of the use of rectified flow on real-world tasks would enhance the justification for the proposed CAF. Can such examples be provided and discussed?

Are there other options to provide the initial velocity field rather than using a pre-trained generative model?

The paper is generally well written. The following are minor typos:
l. 20  a huge computation burden -> a large computational burden
l. 128 thet -> the
l. 176 Experiment -> Experiments or Experimental Evaluation
l. 188 128-dimensional dimensions

**Limitations:**

The limitations section should be moved from the appendix into the main paper as the computational impact of CAF over rectified flow is essential.

---

> ### Author Rebuttal · Authors · 2024-08-07
>
> We highly appreciate the reviewer's effort for the detailed feedback and for spotting the typos. We will make corrections in the final version.
>
> **Q.1 [Significance of work & real-world application]**
>
> **Response to Q.1**
> In response to the concern regarding the significance of our method, we would like to emphasize three key points:
> - **[Fast sampling]:** To address the slow inference of diffusion models, there have been efforts to reduce inference steps at the cost of additional training, such as CM and CTM. Our work shares this motivation, effectively addressing slow inference without compromising quality. We believe our work has substantial potential to make a generative model both fast and accurate, enhancing its applicability.
> - **[Efficient editing]:** Inversion has been an essential technique for real-world applications such as image/video editing [1,2]. However, current methods require 25-100 steps for accurate inversion, whereas our method can significantly reduce the inference time by a few-step inversion. We demonstrate this in two additional tasks: *reconstruction* and *box inpainting*. For the results, please refer to **4. [Inversion & Zero-shot image editing]** in the "**Author Rebuttal**" above.
> - **[ImageNet 64x64 results]:** For real-world examples, we provide additional results on ImageNet 64x64. These results demonstrate the broader applicability of our framework. For the quantitative results, please refer to **1. [ImageNet 64x64 results]** in the “**Author Rebuttal**” above.
>
> ---
> **Q.2 [Impact of pre-trained model]**
>
> **Response to Q.2**
> As the reviewer commented, the performance of CAF can be influenced by the quality of the pre-trained model. This phenomenon is common in other distillation methods, where the student's performance is bounded by the teacher. To address this, we have incorporated auxiliary adversarial loss with real data. This auxiliary loss can be considered as the divergence between the real data distribution and the learned distribution, helping the model surpass the performance of the pre-trained model. Our empirical results in the paper also support this, where CAF’s performance with distillation (FID 1.7) surpasses EDM (FID 2.01).
>
> ---
> **Q.3 [Why constant acceleration is better than constant velocity?]**
>
> **Response to Q.3**
> CAF employs 2nd-order momentum by incorporating acceleration (2nd derivative of position w.r.t time). In contrast, constant velocity flow relies on 1st-order momentum, representing only the velocity. The inclusion of the 2nd-order term in CAF enables it to account for the time-varying nature of velocity, providing a more expressive approximation of the dynamics. This is analogous to higher-order terms in a Taylor series expansion, leading to lower numerical errors during the sampling phase. The superiority of higher-order schemes in reducing numerical errors is well-documented in numerical analysis literature (see LeVeque, 2002, Finite Volume Methods for Hyperbolic Problems).
>
> ---
> **Q.4 [Contribution of reflow and NFSS]**
>
> **Response to Q.4**
> Thank you for your feedback. We would like to clarify that we do *not* intend to claim credit for the reflow procedure or the measurement of straightness, both of which were introduced in the rectified flow. In our paper, we have provided appropriate references and discussions of these concepts in the abstract, introduction, and preliminary sections to acknowledge their original sources and their use in our work. Our primary contributions lie in the development of the CAF framework, which leverages these established techniques to achieve and measure improved performance. We will ensure that our references and discussions are appropriately highlighted to avoid any misunderstanding.
>
> ---
> **Q.5 [Details of CIFAR10]**
>
> **Response to Q.5**
> We will include the details in the revised paper. For training, we generated 12 million pairs from a pre-trained EDM model using 10 class labels. We do not use class labels for unconditional training. For class conditional training, the class labels are transformed into one-hot class embeddings and then element-wise added to the time-step embeddings.
>
> ---
> **Q.6 [Additional training efforts for the pre-trained model]**
>
> **Response to Q.6**
> We acknowledge that the use of a pre-trained model should be considered for the required computational effort during training. To clarify this, we have considered the following cases:
> - **[With pre-trained models]:** In scenarios where pre-trained models are readily available, leveraging them can significantly reduce the overall training time and computational resources required.
> - **[Without pre-trained model]:** In scenarios where pre-trained models are not available, our CAF can be trained from scratch to serve as a model to generate deterministic couplings. To show this, we conducted experiments where CAF was trained solely from real data without any pre-trained models. For the results, please refer to section 2. [Training without pre-trained model] in the "Author Rebuttal". This approach increases the computational efforts but effectively addresses the scenario where pre-trained models are not available.
>
> We will include these discussions in the revised version of our paper to clarify the required computational effort during training.
>
> ---
> **Q.7 [Real-world applications]**
>
> **Response to Q.7**
> Please refer to our response to **Q.1 [Significance of work & real-world application].**
>
> ---
> **Q.8 [Other options to provide the initial velocity]**
>
> **Response to Q.8**
> Thank you for pointing out possible directions for future study. We believe that improving generation quality in few-step regimes without relying on reflow is a promising direction for future research.
>
> ---
> [1] Mokady, Ron, et al, Null-text inversion for editing real images using guided diffusion models, CVPR2023 \
> [2] Huberman-Spiegelglas, Inbar, et al, An edit friendly DDPM noise space: Inversion and manipulations, CVPR2024

---

> > ### Comment · Reviewer_oUYA · 2024-08-09
> > **Rebuttal well-addressed my concerns**
> >
> > Thank you for the detailed rebuttal.
> > Based on the explanations and especially the demonstrations of additional applications, I have revised my recommendation to weak accept.

---

> > > ### Author Response · Authors · 2024-08-11
> > >
> > > We greatly appreciate the reviewer’s response and are truly thankful for the constructive feedback that has helped enhance our work. Thank you for dedicating your time and effort to providing us with such valuable insights.

---

### Official Review · Reviewer_rYLJ · 2024-07-10

**Soundness:** 2
**Presentation:** 2
**Contribution:** 3
**Rating:** 4
**Confidence:** 4

**Summary:**

This work proposes Constant Acceleration Flow (CAF), which, instead of learning a velocity-based flow model like in Flow Matching, jointly trains an initial velocity model together with a constant time-dependent acceleration model. This framework aims to learn straighter paths and thus enable better results for few-step generation.

Additionally, they propose to condition the acceleration model on the initial velocity and propose to improve the initial velocity using a reflow procedure. CAF is empirically validated on toy data and CIFAR-10, including ablations on the different elements of the proposed framework.

**Strengths:**

- The Constant Acceleration Flow is a new formulation for achieving straighter trajectories in flow-based generative models derived based on the assumption of constant acceleration.
- Good empirical results on CIFAR-10
- Includes important ablations on the velocity conditioning as well as on the magnitude of the initial velocity

**Weaknesses:**

"Reflow" for initial velocity:
- The reflow procedure was proposed to straighten the paths of an existing generative model. The authors propose to use a "reflow" procedure for training CAF. However, they don't generate the couplings with a model trained with CAF but instead with a pre-trained EDM model, which is very different from how the reflow procedure was proposed. Additionally, it is unclear how dependent CAF is on this procedure and, thus, from the pre-trained generative model. In the CIFAR-10 ablation, the setting with constant acceleration and velocity conditioning but no "reflow" procedure (and thus no pre-trained generative model) is crucial to answer the question of whether CAF also works without a pre-trained model. If it does not, then I would argue that the proposed framework is more of a new distillation technique rather than a new generative model. This contextualization should also be more clearly explained in the main text.

Missing discussion and contextualization of related work:
- Acceleration Generative Modeling has been proposed in [1]. While CAF is based on Flow Matching and a constant velocity is chosen, [1] considers Bridge Matching in a stochastic optimal control framework with a changing velocity induced by the acceleration prediction. However, their trained model is also a parameterized acceleration prediction that takes as input the current data point $x_t$, current time $t$, and the current velocity $v_t$ effectively conditioning on the velocity. This framework is very related to the approach proposed by the authors, and the connection should be discussed in detail in the main paper. Additionally, an experimental comparison illustrating the differences between the two approaches could be beneficial.
- Coupling preservation through initial velocity conditioning: The authors refer to the reflow approach as existing related work. However, the reflow approach was proposed to straighten paths, not preserve couplings. Moreover, the problem of coupling preservation has been thoroughly analyzed in [2], which proposes to use a source point conditioning, i.e. conditioning on $x_0$. How does Augmented Flow/Bridge Matching (Flow/Bridge Matching with $x_0$ conditioning) compare to CAF? As mentioned, velocity conditioning is also used in [1].
- A possible way to empirically compare to these competing methods would be to extend the ablation study on CIFAR-10 to include [1] and [2].

Minor Weaknesses:
- Experiments are only conducted on CIFAR-10. Including at least one more dataset, e.g. 64x64 ImageNet/CelebA/etc, could strengthen the empirical results.

[1] Tianrong Chen and Jiatao Gu and Laurent Dinh and Evangelos A. Theodorou and Joshua Susskind and Shuangfei Zhai. "Generative Modeling with Phase Stochastic Bridges". In ICLR 2024.

[2] Valentin De Bortoli and Guan-Horng Liu and Tianrong Chen and Evangelos A. Theodorou and Weilie Nie. "Augmented Bridge Matching". In Arxiv 2023.

**Questions:**

- As mentioned above, how dependent is CAF on the pre-trained model? Does it also work well without any pre-trained model?
- Does the 2-Rectified Flow comparison for the CIFAR-10 experiments also use generated samples by EDM for its first training round?
- Can the velocity and acceleration models be trained separately? E.g. first train the initial velocity model and then train the acceleration model?

**Limitations:**

- Again, how dependent is CAF on the pre-trained model?
- Training both an acceleration and a velocity prediction model effectively doubles the amount of total parameters. This could be mentioned as another limitation.

---

> ### Author Rebuttal · Authors · 2024-08-06
>
> We truly appreciate the reviewer for the constructive feedback. We would like to clarify the reasoning behind our approach and address the concerns raised.
>
> **Q.1** **[Why use EDM for reflow?]**
>
> **Response to Q.1**
> In response to the reviewer’s concern regarding the reflow procedure, we would like to emphasize that the reflow procedure can be flexibly applied to any pre-trained diffusion model, which is a notable advantage of reflow, as demonstrated in InstaFlow [1]. Based on these findings and the inherent flexibility of the reflow procedure, we utilized the pre-trained EDM model to ensure a fair comparison against other fast sampling models, such as CM and CTM, which also utilized pre-trained EDM models.
>
> ---
> **Q.2 [Contextualization of work]**
> The proposed framework seems to be more of a new distillation technique rather than a new generative model. This contextualization should also be more clearly explained in the main text.
>
> **Response to Q.2** We acknowledge the importance of contextualizing the method. We will include the below discussions and contextualize our work more clearly in the main text of the final version. To address the reviewer’s concern comprehensively, we have considered two key points:
>
> - **[Fast sampling as our main motivation]**: We would like to emphasize that our primary focus is to propose an ODE framework based on Rectified Flow that enables fast generation with high accuracy (as stated in the abstract, introduction, and preliminary sections), rather than introducing a new generative model family. To achieve this goal, we have proposed three techniques: constant acceleration, initial velocity conditioning (IVC), and reflow. Each of these techniques independently improves the estimation accuracy for single-step generation, as demonstrated in Table 2 of our paper.
> - **[CAF as an independent framework]**: To address the reviewer’s concern regarding the reliance on reflow, we conducted experiments where CAF was trained solely from real data (CIFAR-10) without the deterministic couplings generated from the pre-trained model (denoted as 1-CAF). For the quantitative results, please refer to section **2. [Training without pre-trained model]** in the "**Author Rebuttal**". The results show that 1-CAF effectively learns the real data distribution without reflow, even outperforming 1-Rectified Flow (RF). However, similar to 1-RF, 1-CAF loses its accuracy in extremely few-step regimes. We believe that improving generation quality in these regimes without the deterministic couplings is an interesting direction for future research.
>
> ---
> **Q.3 [Missing discussion with AGM]**
>
> **Response to Q.3** Thank you for bringing out the related work that we have missed. We found that AGM elegantly formulates a new generative model using the acceleration term based on SOC theory. Since this concurrent work was published a few weeks before our submission, we unfortunately missed this important contribution. We will ensure that discussions of AGM and related works are included in the revised paper.
>
> The main difference between AGM and CAF is that CAF assumes constant acceleration, whereas AGM predicts time-dependent acceleration. Our constant acceleration framework leads to a simpler sampling procedure (Eq. 11 in our paper) based on an analytically closed-form solution (Eq. 5 in our paper). This closed-form solution enables few-step (N<3) sampling with high accuracy when learned with deterministic couplings. To demonstrate this, we additionally compare the generation results of AGM and CAF on CIFAR-10. For the quantitative results, please refer to section **3. [Comparison with AGM]** in the "**Author Rebuttal**".  The result demonstrates that CAF outperforms AGM in terms of FID scores in an extremely few-step regime. Moreover, we provide qualitative comparisons between CAF and AGM in the attached PDF file, where CAF generates images with more high-frequency details than AGM for few-step sampling. These results highlight the distinct advantages of our framework over AGM, particularly in terms of single-step sampling accuracy.
>
> ---
> **Q.4 [Missing discussion with AugBM]**
>
> **Response to Q.1**
> Thank you for pointing out the related work that we have missed. We recognize that AugBM shares a similar motivation with our method in preserving couplings and addressing flow (bridge) crossing by conditioning auxiliary information to the network. However, while AugBM aims to improve coupling preservation specifically for image-to-image translation tasks, our main contribution lies in developing an ODE framework that enables fast sampling with high accuracy. We will include a detailed discussion of AugBM and related works in the revised paper to provide a more comprehensive comparison.
>
> ---
> **Q.4 [Training setting of 2-Rectified Flow]**
>
> **Response to Q.4**
> Yes, in our experiments, we used the same dataset generated by the pre-trained EDM for both CAF and RF.
>
> ---
> **Q.5 [Separate training]**
>
> **Response to Q.5**
> Thank you for the insightful suggestion. We investigated the possibility of training the velocity and acceleration models separately. In our experiments on the CIFAR-10 conditional setting, we found that training each network separately led to more stable training dynamics. This approach resulted in a slight improvement in the FID score, reducing it from 4.18 to 3.32.
>
> ---
> **Q.6 [Computational cost]**
>
> **Response to Q.6** We agree with the reviewer that our framework requires additional learnable parameters for both acceleration and velocity prediction models. This can be a limitation, particularly for low-resource systems. One potential mitigation strategy could be training a single model to learn both velocity and acceleration by conditioning the model appropriately. We will include this discussion in the limitations section of the revised paper.
>
> ---
> [1] Liu, Xingchao, et al. Instaflow: One step is enough for high-quality diffusion-based text-to-image generation, ICLR 2024

---

> > ### Comment · Reviewer_rYLJ · 2024-08-12
> >
> > Thank you for the detailed answers and added experiments.
> >
> > > ***The result demonstrates that CAF outperforms AGM in terms of FID scores in an extremely few-step regime.***
> >
> > For the rebuttal, the authors compare CAF with reflow vs AGM without reflow. I believe the contribution of the [i] constant acceleration flow should be disentangled from the [ii] use of reflow. [ii] Reflow should also significantly improve the few-step results of AGM. To fairly compare the two frameworks, a comparison of just [i] vs AGM is needed or/and a comparison of [i] + [ii] vs AGM + [ii]. In my opinion, the reported results in the author rebuttal do not constitute a fair comparison between AGM and CAF.
> >
> > > ***This closed-form solution enables few-step (N<3) sampling with high accuracy when learned with deterministic couplings.***
> >
> > Is there an intuition of why this closed-form enable few-step with deterministic coupling over AGM with deterministic couplings? As mentioned above, this should also be confirmed empirically.
> >
> > > ***Since this concurrent work was published a few weeks before our submission.***
> >
> > AGM was uploaded to Arxiv October 2023, which does not make it concurrent given the NeurIPS Contemporaneous Work guidelines but previous work.

---

> ### Author Response · Authors · 2024-08-13
>
> Thank you for the response. We'd like to clarify the additional questions the reviewer raised.
>
> > ***AGM was uploaded to Arxiv October 2023, which does not make it concurrent given the NeurIPS Contemporaneous Work guidelines but previous work.***
>
> First, we would like to clarify and sincerely apologize for any confusion: at the time of our submission, we were *genuinely unaware* of the AGM. As mentioned in our previous response, we fully recognize the AGM's contribution to the field of acceleration modeling and will ensure that discussions of AGM and related works are included in the revised paper.
>
> > ***Is there an intuition of why this closed-form enables few-step with deterministic coupling over AGM with deterministic couplings?***
>
> Since the CAF ODE assumes that the acceleration term is *constant* with respect to time, there is no need to iteratively solve complex time-dependent differential equations. This simplification allows for a direct closed-form solution that supports efficient and accurate sampling in just a few steps, given that the learned velocity and acceleration models are sufficiently accurate.
>
> In contrast, AGM’s acceleration term is *time-varying*. This variability means that the differential equation cannot be simplified in the same way, often necessitating multiple iterative steps to approximate the true solution accurately. The constant acceleration assumption in CAF is similar to the concept of rectified flow (constant velocity), which is known to make solving differential equations more efficient than in diffusion models.
>
> Below are the details:
>
> In CAF ODE (equation 4 in the paper), the solution for the final sample is given by:
>
> $x_1 =x_0 +\int_0^1v(x_0,0)+a(x_t,t)tdt=x_0+v(x_0,0) +\int_0^1a(x_t,t)tdt$
>
> Thanks to the constant acceleration assumption, the integral simplifies to:
>
> $x_1=x_0+v(x_0,0)+ a(x_0,0)\int_0^1tdt=x_0+v(x_0,0)+ \frac{1}{2}a(x_0,0)$
>
> This result corresponds to the one-step sampling version of equation 11 in our paper.
>
> > ***For the rebuttal, the authors compare CAF with reflow vs AGM without reflow. I believe the contribution of the [i] constant acceleration flow should be disentangled from the [ii] use of reflow. [ii] Reflow should also significantly improve the few-step results of AGM. To fairly compare the two frameworks, a comparison of just [i] vs AGM is needed or/and a comparison of [i] + [ii] vs AGM + [ii]. In my opinion, the reported results in the author rebuttal do not constitute a fair comparison between AGM and CAF.***
>
> Thank you for your valuable feedback. We understand your concern regarding the need to disentangle the use of the reflow.
>
> First, we would like to clarify the rationale behind our comparisons. In our proposed method, the reflow procedure is an essential component designed to achieve an accurate solution of our CAF ODE. AGM, on the other hand, does not incorporate reflow in its original methodology. In our rebuttal, we have fully utilized the AGM method in its **standard form** as proposed by the authors, using their official code without making any alterations. This approach reflects the typical use cases and capabilities of both methods as they were originally intended to be used.
>
> To address the reviewer's concern, we conducted additional experiments where AGM was trained with deterministic couplings generated from EDM (the same as our reflow setting). We replaced $\epsilon_0$ and $x_1$ in AGM with noise-image pairs from EDM and followed the experimental setup in the official AGM code. The results are summarized in the table below:
> |                        | N |   FID $\downarrow$   |
> |------------------------|:-:|:-----:|
> | AGM-ODE without reflow | 5 | 11.88 |
> | AGM-ODE with reflow    | 5 |  15.23 |
> | CAF                    | 1 |  5.15 |
>
> In fact, incorporating reflow into AGM did not necessarily improve its performance in the few-step regime. We assume that this is because AGM may require noise-velocity-image triplets. This indicates that effectively integrating the reflow mechanism within the AGM framework is not straightforward with a pre-trained diffusion model, and it requires extra effort to build these triplets that go beyond the scope of our current comparison.
>
> ---
> We hope our explanations address the reviewer’s concerns regarding AGM.

---

> ### Author Response · Authors · 2024-08-14
>
> Dear Reviewer rYLJ,
>
> We sincerely appreciate your dedication and constructive feedback on our work. As the reviewer-author discussion period ends soon, we would like to kindly ask if you have any remaining concerns or questions.
>
>
> Best regards, \
> Authors

---

> > ### Comment · Reviewer_rYLJ · 2024-08-14
> >
> > Thanks for the further explanations and results. This further clarified my concerns and I've decided to raise my score 3 -> 4.

---

### Official Review · Reviewer_GaYE · 2024-07-12

**Soundness:** 3
**Presentation:** 3
**Contribution:** 3
**Rating:** 6
**Confidence:** 5

**Summary:**

This paper develops a new flow based generative model whose vector field is constructed similar to rectified flows (straight path between source and target samples), but instead of using a path with a constant speed, uses a path with constant acceleration.  This requires learning a neural network to parametrize the initial velocity field and another neural network to parametrize the acceleration field.  To improve their model's performance, the authors also propose parametrizing the acceleration neural network using the initial velocity and also applying reflows to the initial velocity.  The proposed method helps mitigate the flow crossing problem at a model level, which should in theory avoid excess reflows to learn non-crossed flows.  The experiments section demonstrates the ability of the proposed method to learn high performing generative models on single step generation on the CIFAR-10 dataset.

**Strengths:**

The idea is novel, solves a relevant problem and fits nicely into the current landscape of the flow based generative modeling research area.  The paper was easy to read and the method itself is simple enough to understand and implement after a single pass over the paper.  The main contribution of the paper, in my opinion, is the conditioning of the acceleration neural network on the initial velocity.  This gives the final trained model a non-markov path measure, which is something that is of current interest to the field (Augmented Bridge Matching, De Bortoli et al., 2023).

**Weaknesses:**

- The related work should include references to building diffusion models with non-markov path measures (see De Bortoli et al., 2023 and its related work).
- For completeness, the paper should include proofs to show that the learned distribution is the same as the data distribution.  This follows directly from interpreting Eq. 2 as a conditional flow matching objective (Lipman et al., 2023).
- In the initial presentation of the vector field $a(x_t,t)$ at the start of section $4.1$, it would make more sense to drop the dependence on $t$ to emphasize that the ground truth acceleration field is constant in time.
- The scope of the empirical evaluation is limited.  The authors should have evaluated on more datasets than just CIFAR-10.  There are other small scale image datasets, like ImageNet-32, that could have been evaluated.

**Questions:**

- Is there a reason why algorithm 1, line 6, updates $\theta$ before computing the loss for $a_\phi$?

**Limitations:**

The authors adequately addressed the limitations.

---

> ### Author Rebuttal · Authors · 2024-08-06
>
> **Q.1 [Missing discussion with AugBM]**  The related work should include references to building diffusion models with non-markov path measures (see De Bortoli et al., 2023 and its related work).
>
> **Response to Q.1**
>
> We appreciate the reviewer for pointing out the related works that we have missed. We recognize that AugBM shares a similar motivation with our method in preserving couplings and addressing flow (bridge) crossing by conditioning auxiliary information to the network.
> However, while AugBM aims to improve coupling preservation specifically for image-to-image translation tasks, our main contribution lies in developing an ODE framework that enables fast sampling with high accuracy. To achieve this, we proposed a flow with constant acceleration that provides a one-step closed-form solution (Eq. 5 in our paper). This closed-form solution enables few-step (N<3) sampling (Eq.11 in our paper) with high accuracy when learned with deterministic couplings, as demonstrated by our experimental results.
>
> We will include a detailed discussion of AugBM and related works in the revised paper to provide a more comprehensive comparison.
>
> ---
> **Q.2 [Proof of marginal preserving]**
> For completeness, the paper should include proofs to show that the learned distribution is the same as the data distribution.
>
> **Response to Q.2**
> We highly appreciate the feedback and acknowledge the importance of demonstrating the marginal preservation property of our approach. To address the reviewer’s concern, we provide a derivation that the flow induced by our Constant Acceleration Flow (CAF) ordinary differential equation (ODE) preserves the marginal of the data distribution, established by the definitions and theorems in [1].
>
> ***Definition 1***: *For a path-wise continuously differentiable process $\mathbf{x} = {\mathbf{x}_t : t \in [0,1]}$, we define its expected velocity $v^{\mathbf{x}}$ and acceleration $a^{\mathbf{x}}$ as follow:*
> $$
> \
> v^{\mathbf{x}}(x,t) = \mathbb{E}\left[\frac{d\mathbf{x}_t}{dt} \ \bigg| \ \mathbf{x}_t = x\right], \quad a^{\mathbf{x}}(x,t) = \mathbb{E}\left[\frac{d^2\mathbf{x}_t}{dt^2} \ \bigg| \ \mathbf{x}_t = x\right], \quad \forall x \in \text{supp}(\mathbf{x}_t).
> \
> $$
> *For $x \notin \text{supp}(\mathbf{x}_t)$, the conditional expectation is not defined, and we set $v^{\mathbf{x}}$ and $a^{\mathbf{x}}$ arbitrarily, for example, $v^{\mathbf{x}}(x,t) = 0$ and $a^{\mathbf{x}}(x,t) = 0$.*
>
> ***Definition 2*** [1]: *We denote that $\mathbf{x}$ is rectifiable if $v^{\mathbf{x}}$ is locally bounded and the solution of the integral equation of the form*
> $$
> \
> \mathbf{z}_t = \mathbf{z}_0 + \int_0^t v^{\mathbf{x}}(\mathbf{z}_t, t) dt, \quad \forall t \in [0,1], \quad \mathbf{z}_0 = \mathbf{x}_0,
> \
> $$
> *exists and is unique. In this case, $\mathbf{z} = {\mathbf{z}_t : t \in [0,1]}$ is called the rectified flow induced by $\mathbf{x}$.*
>
> ***Theorem 1*** [1]: *Assume $\mathbf{x}$ is rectifiable and $\mathbf{z}$ is its rectified flow. Then, $\text{Law}(\mathbf{z}_t) = \text{Law}(\mathbf{x}_t)$ for all $t \in [0,1]$.*
>
> \
> Refer to [1] for the proof of ***Theorem 1***. We will now show that our CAF ODE satisfies ***Theorem 1*** by proving that our proposed ODE induces $\mathbf{z}$, which is the rectified flow as defined in ***Definition 2***. In our paper, we defined the CAF ODE as
> $$
> \frac{d\mathbf{x}_t}{dt} = \left. \frac{d\mathbf{x}_t}{dt} \right|{t=0} + \frac{d^2\mathbf{x}_t}{dt^2} \cdot t.
> $$
> By taking the conditional expectation on both sides, we obtain
> $$
> v^{\mathbf{x}}(x,t) = v^{\mathbf{x}}(x,0) + a^{\mathbf{x}}(x,t) \cdot t,
> $$
> from ***Definition 1***. Then, the solution of the integral equation of CAF ODE is identical to the solution in ***Definition 2***:
> $$
> \mathbf{z}_t = \mathbf{z}_0 + \int_0^t \left( v^{\mathbf{x}}(\mathbf{z}_0, 0) + a^{\mathbf{x}}(\mathbf{z}_t, t) \cdot t \right) dt \
> = \mathbf{z}_0 + \int_0^t v^{\mathbf{x}}(\mathbf{z}_t, t) dt.
> $$
> This indicates that $\mathbf{z}$ induced by CAF ODE is also a rectified flow. Therefore, CAF ODE satisfies the marginal preserving property, i.e., $\text{Law}(\mathbf{z}_t) = \text{Law}(\mathbf{x}_t)$, as stated in ***Theorem 1***.
>
> ---
> **Q.3 [Expression of acceleration model]**
> In the initial presentation of the acceleration field $a(x_t,t)$ at the start of section 4.1, it would make more sense to drop the dependence on $t$ to emphasize that the ground truth acceleration field is constant in time.
>
> **Response to Q.3**
>
> Thank you for your suggestion. We agree that emphasizing that the ground truth acceleration field is constant in time can enhance clarity. We will consider revising the notation.
>
> ---
> **Q.4 [Limited evaluation]**
> The scope of the empirical evaluation is limited. The authors should have evaluated on more datasets than just CIFAR-10. There are other small-scale image datasets, like ImageNet-32, that could have been evaluated.
>
> **Response to Q.4**
>
> Thank you for your feedback regarding the scope of our empirical evaluation. To address the reviewer’s concern, we have extended our evaluation to include additional generation results on ImageNet 64x64. These results demonstrate the broader applicability and effectiveness of our framework beyond CIFAR-10. For the quantitative results, please refer to **1. [ImageNet 64x64 results]** in the “**Author Rebuttal**” section above.
>
> ---
> **Q.5 [Training algorithm]** Is there a reason why algorithm 1, line 6, updates  $\theta$ before computing the loss for $a_\phi$?
>
> **Response to Q.5**
>
> Thank you for your question. In our experiments, we found that updating the initial velocity model before computing the loss for the acceleration model leads to more stable training dynamics.
>
> ---
> [1] Liu, Xingchao, Chengyue Gong, and Qiang Liu, Flow straight and fast: Learning to generate and transfer data with rectified flow, ICLR 2023

---

> > ### Comment · Reviewer_GaYE · 2024-08-08
> >
> > Thank you for the response and for the updated experiments.  My opinion on the paper is the same so I am going to keep the same score.

---

> > > ### Author Response · Authors · 2024-08-11
> > >
> > > We sincerely appreciate the reviewer’s prompt response and are grateful for the constructive comments that have strengthened our work. Thank you for dedicating your time and effort to providing us with such valuable insights.

---

### Author Rebuttal · Authors · 2024-08-06

We sincerely appreciate all reviewers' detailed feedback. Here, we present additional quantitative results to address their concerns.

---
**1. [ImageNet 64x64 Results]**

In response to the concerns about the limited evaluation of our method, we provide additional quantitative results (FID, Inception Score, and Recall) on ImageNet 64x64. The table below summarizes the results:
| Model  |  N  |  FID $\downarrow$  |   IS $\uparrow$  | Recall $\uparrow$ |
|----------------------------------------------------|:---:|:-----:|:-----:|:------:|
| _GAN Models_     |     |       |       |        |
| BiGGAN-deep    |  1  |  4.06 |   -   |  0.48  |
| StyleGAN-XL    |  1  |  2.09 | **82.35** |  0.52  |
|       |     |       |       |        |
| _Diffusion Model / Consistency Model_    |     |       |       |        |
| ADM     | 250 |  2.07 |   -   |  **0.63**  |
| EDM   |  79 |  2.44 | 48.88 |  0.67  |
| iCT-deep  |  2  |  2.77 | - |  0.62  |
| iCT-deep  |  1  |  3.25 | - |  **0.63**  |
|                                                    |     |       |       |        |
| _Diffusion Models - Rectified flow_                |     |       |       |        |
| **CAF (Ours)**   |  1  |  9.29 | 42.73 |  0.627 |
|                                                    |     |       |       |        |
| _Diffusion Models - Distillation_                  |     |       |       |        |
| CD  |  1  |  6.2  | 40.08 |  **0.63**  |
| Diff-Instruct  |  1  |  5.77  | - |  -  |
| CTM  |  2  |  1.73 | 64.29 |  0.57  |
| CTM  |  1  |  1.92 | 70.38 |  0.57  |
|
| _Diffusion Models - Rectified Flow + Distillation_ |     |       |       |        |
| **CAF (+distill + GAN) (Ours)**    |  1  |  **1.69** | 62.03 |  0.621 |

These results demonstrate that our method achieves comparable performance compared to recent state-of-the-art models, highlighting the generalization capability of our approach to large-scale datasets. Additionally, we have included the ImageNet generation results from our trained model with a single step in **Figure 4** of the attached PDF file.

---
**2. [Training without pre-trained model]**

To address the reviewer’s concern regarding the reliance on a pre-trained model, we report additional results where CAF was trained solely from real data (CIFAR-10) without the deterministic couplings generated by the pre-trained model (denoted as 1-CAF).
| Model | N |  FID $\downarrow$ |
|-------|:---:|:----:|
| 1-RF  |  1  |   335   |
|   **1-CAF**  |  1 |  328 |
| 1-RF |  70  |   4.7   |
|  **1-CAF**  |  70 | **3.53** |

The results show that 1-CAF effectively learns the real data distribution without the deterministic couplings, even outperforming 1-Rectified Flow (RF). However, similar to 1-RF, we lose its accuracy in extremely few-step regimes. We believe that improving quality in these regimes without the deterministic couplings is an interesting direction for future research.

---
**3. [Comparison with AGM]**

As the reviewer rYLJ requested, we additionally compare the generation results of Acceleration Generative Model (AGM) [1] and CAF in the table below.
| Dataset        | Model      |  N |  FID $\downarrow$  |
|----------------|------------|:--:|:-----:|
| CIFAR-10       | AGM-ODE    |  50 | 2.46 |
|    | AGM-ODE |  5 |  11.88 |
|    | **CAF (Ours)** |  1 |  **1.7** |
| ImageNet 64x64 | AGM-ODE    | 30 | 10.07 |
|    | AGM-ODE    | 20 | 10.55 |
|    | **CAF (Ours)** |  1 |  **1.69** |

The results demonstrate that CAF outperforms AGM in terms of FID scores in an extremely few-step regime. Moreover, we provide qualitative comparisons between CAF and AGM in **Figure 3** of the attached PDF file, where CAF generates images with more high-frequency details than AGM for few-step sampling. These results highlight the distinct advantages of CAF over AGM, particularly in terms of few-step sampling accuracy. For a detailed discussion, please refer to our response to the revierwer rYLJ **Q.3 [Missing discussion with AGM]**.

---
**4. [Inversion & Zero-shot image editing]**

As the reviewer oUYA requested, we demonstrate our CAF's capability on real-world applications by conducting zero-shot tasks on CIFAR-10 test datasets (*reconstruction* and *box inpainting*). Since our framework is based on ODEs, we can solve our CAF ODE in reverse to perform inversion like DDIM [2]. For reconstruction, we followed the procedure as DDIM. For box inpainting, we inject conditional information (non-masked image region) into the iterative inversion and reconstruction procedure. As demonstrated in the table below, we achieve better reconstruction quality and zero-shot inpainting capability even in a few steps compared to the baselines. This improvement is due to the superior coupling preservation capability of CAF (in Table 4 of our paper). Moreover, we provide qualitative results in **Figure 1 and 2** of the attached PDF file that align with the quantitative results.

|  Reconstruction | N |  PSNR $\uparrow$ | LPIPS $\downarrow$ |
|-------------|:-:|:-----:|:-----:|
| CM  | - |  N/A  |  N/A  |
| CTM  | - |  N/A  |  N/A  |
| EDM  | 4 | 13.85 | 0.447 |
| 2-RF  | 1 | 29.33 | 0.204 |
| **CAF (Ours)** | 1 | **30.27** | **0.171** |

| Box inpainting | N |  FID $\downarrow$  |
|----------------|:---:|:-----:|
| CM  |  18 | 13.16 |
| 2-RF  |  10  | 16.41 |
| **CAF (Ours)** |  10  |  **9.79** |

These results demonstrate that our method can significantly reduce the inference time required for recent methods that utilize inversion for various real-world applications such as image/video editing.

---
[1] Chen, Tianrong, et al, Generative modeling with phase stochastic bridges, ICLR 2024 \
[2] Song, Jiaming, Chenlin Meng, and Stefano Ermon, Denoising diffusion implicit models, ICLR 2021

---

### Decision · Program_Chairs · 2024-09-25

**Decision:**

Accept (poster)

**Comment:**

This paper received two weak acceptances and one borderline rejection. Reviewer rYLJ generally agreed with the two positive reviews but raised a key concern regarding the concurrent work AGM [1], which might have a similar approach to the acceleration generation modeling framework. In the rebuttal, the authors provided a comparison with AGM, highlighting differences in the acceleration strategy and the closed-form solution. The comparison results also further verified the effectiveness. These points effectively justify the novelty of this work, especially since constant acceleration is a central claim in the title. Consequently, the AC agreed that the work retains its novelty despite the concurrent work on arXiv earlier, which had not been fully published before the NeurIPS submission deadline.

Additionally, the AC considered the generalization of this approach to other datasets, as noted by Reviewer oUYA. The rebuttal included further experiments on ImageNet 64x64, which strengthened the justification and made the work more convincing.

Based on the novelty, solid experiments presented in the paper, and additional evaluations in the rebuttal, the AC concurs with the majority of reviewers to accept the paper. For the camera-ready version, it is suggested to add a subsection discussing AGM and include this work in the related works section. Furthermore, it is encouraged to add generalization results on other datasets in the final version.